# Efficient Model-Based Multi-Agent Mean-Field Reinforcement Learning

**Barna Pásztor**                                      *barna.pasztor@ai.ethz.ch*
*ETH Zürich*

**Ilija Bogunovic**                                      *i.bogunovic@ucl.ac.uk*
*University College London*

**Andreas Krause**                                      *krausea@ethz.ch*
*ETH Zürich*

**Reviewed on OpenReview:** *https://openreview.net/forum?id=gvcDSDYUZx*

## Abstract

Learning in multi-agent systems is highly challenging due to several factors including the non-stationarity introduced by agents' interactions and the combinatorial nature of their state and action spaces. In particular, we consider the *Mean-Field Control* (MFC) problem which assumes an asymptotically infinite population of identical agents that aim to collaboratively maximize the collective reward. In many cases, solutions of an MFC problem are good approximations for large systems, hence, efficient learning for MFC is valuable for the analogous discrete agent setting with many agents. Specifically, we focus on the case of *unknown* system dynamics where the goal is to simultaneously optimize for the rewards and learn from experience. We propose an efficient *model-based* reinforcement learning algorithm, $M^3$–UCRL, that runs in episodes, balances between exploration and exploitation during policy learning, and *provably* solves this problem. Our main theoretical contributions are the first general regret bounds for model-based reinforcement learning for MFC, obtained via a novel mean-field type analysis. To learn the system's dynamics, $M^3$–UCRL can be instantiated with various statistical models, e.g., neural networks or Gaussian Processes. Moreover, we provide a practical parametrization of the core optimization problem that facilitates gradient-based optimization techniques when combined with differentiable dynamics approximation methods such as neural networks.

## 1 Introduction

Multi-Agent Reinforcement Learning (MARL) extends the scope of reinforcement learning (RL) to multiple agents acting in a shared system. It is receiving considerable attention given a great number of real-world applications including autonomous driving (Shalev-Shwartz et al., 2016), finance (Lee et al., 2002; Lee et al., 2007; Lehalle & Mouzouni, 2019), social science (Castelfranchi, 2001; Leibo et al., 2017), swarm motion (Almulla et al., 2017), e-sports (Vinyals et al., 2019; Pachocki et al., 2018), and traffic routing (El-Tantawy et al., 2013), to name a few. Despite the recent popularity, analyzing agents' performance in these systems remains a particularly challenging task for several reasons including non-stationarity, scalability, competing learning goals, and varying information availability of agents. In this work, we target the challenges of *non-stationarity* and *scalability*. From the perspective of an individual agent, the dynamics of the system are non-stationary, since other agents update their behaviour, which alters over time how the environment reacts to certain actions of the agent. Multi-agent systems also suffer from a *combinatorial nature* due to the exponential growth of the state and action space as the number of agents in the system increases.

To tackle the previous challenges, *Mean-Field Control* (MFC) exploits the insight that many relevant MARL problems involve a large number of *very similar* agents that are coordinated via a centralized controller.

As a motivating application, consider a ride-hailing service in which a central dispatcher coordinates the routes of many drivers around the city. MFC considers the limiting regime of controlling *infinitely many identical collaborative* agents, and utilizes the notion of mean-field approximation from statistical physics (Weiss, 1907) to model their interactions. Specifically, instead of focusing on the individual agents and their interactions, the state of the system is described via the agents' state distribution and the dynamics and rewards are formulated accordingly. Despite the introduced approximations, the solution to the MFC problem often remains adequate for the finite agent equivalent (Lacker, 2017).

The focus of this work is on learning the optimal policies in MFC when the underlying dynamics are *unknown*. In complex domains such as fleets of autonomous vehicles or robot swarms, this is often the case, and the models consequently need to be learnt from data, giving rise to a well-known exploration–exploitation dilemma. In this paper, we propose the *Model-Based Multi-Agent Mean-Field Upper-Confidence RL* algorithm ($\mathrm{M}^3$–UCRL) that is provably efficient and can effectively trade-off between exploration and exploitation during policy learning. Similarly to other Model-Based Reinforcement Learning (MBRL) algorithms, $\mathrm{M}^3$–UCRL collects data about the unknown dynamics online (i.e., by proposing and executing a policy in the true system) and estimates the possible dynamics the system might follow. To optimize the agent's actions, it simulates the unknown system by using the estimated dynamics. Chua et al. (2018) has shown that MBRL can solve challenging single-agent RL problems with low sample complexity. We transfer this property to multi-agent problems and show that $\mathrm{M}^3$–UCRL can efficiently solve the MFC problem with unknown dynamics.

**Related work.** Our $\mathrm{M}^3$–UCRL extends the class of Model-Based Reinforcement Learning (MBRL) algorithms that rely on the *optimism-in-the-face-of-uncertainty* principle. Algorithms based on this approach achieve optimal regret for tabular MDPs (Brafman & Tennenholtz, 2002; Auer et al., 2009; Efroni et al., 2019; Domingues et al., 2020), while in the continuous setting, Abbasi-Yadkori & Szepesvári (2011) and Jin et al. (2020) show $\tilde{\mathcal{O}}(\sqrt{T})$ regret bounds for LQR and linear MDPs, respectively. The recently designed episodic model-based algorithms (Chowdhury & Gopalan, 2019; Kakade et al., 2020; Curi et al., 2020; Sessa et al., 2022) achieve similar regret in the kernelized setting. Sessa et al. (2022) consider the general multi-agent setting, and prove a regret bound with $\mathcal{O}(N^{H/2})$ dependence on the number $N$ of agents. In comparison, $\mathrm{M}^3$–UCRL assumes an asymptotically large population of identical agents. Crucially, as we will show, its performance does *not* depend on the number of agents, making it more suitable for large systems.

*Multi-Agent Reinforcement Learning (MARL)* has seen tremendous progress in recent years as many real-world applications involve interactions among a number of agents. Buşoniu et al. (2008) provide an overview of classical and early MARL results. Several other MARL surveys focus on: non-stationarity induced by the agent interactions (Hernandez-Leal et al., 2017), deep MARL (Hernandez-Leal et al., 2019; Nguyen et al., 2020), theoretical foundations in Markov/stochastic/extensive-form games (Zhang et al., 2019) and cooperative MARL (OroojlooyJadid & Hajinezhad, 2019). Recently, Zhang et al. (2019) and Nguyen et al. (2020) recognize the lack of practical model-based MARL algorithms (with an early exception of the R-Max algorithm proposed by Brafman & Tennenholtz (2000; 2002) for two-player zero-sum Markov Games). We fill this gap by designing a practical model-based algorithm that considers a large population of agents, and that is compatible with deep models.

Modeling interaction via the mean-field approach in MARL has recently gained popularity due to the introduction of Mean-Field Games (MFG) (Lasry & Lions, 2006a;b; Huang et al., 2006; 2007). MFG consider the limiting regime of a competitive game among identical agents with the objective of reaching a Nash Equilibrium. Various works (Yin et al., 2010; 2013; Cardaliaguet & Hadikhanloo, 2017; Guéant et al., 2011; Bensoussan et al., 2013; Carmona & Delarue, 2013; Carmona et al., 2018; Gomes et al., 2014) mainly consider solving MFGs in continuous-time, while some very recent ones (Yang et al., 2017; 2018; Fu et al., 2020; Guo et al., 2019; 2020; Elie et al., 2020; Agarwal et al., 2019) analyze the problem from the discrete-time RL perspective. The progress of the field has been recently summarized by Laurière et al. (2022).

At the same time, cooperative discrete-time *Mean-Field Control*, the main focus of this paper, has received significantly less attention. Gast et al. (2012); Motte & Pham (2019); Gu et al. (2020; 2021) and Bäuerle (2021) consider planning with known dynamics. In the learning setting, Wang et al. (2020); Carmona et al. (2019b); Gu et al. (2021); Angiuli et al. (2022; 2021) extend the Q-Learning algorithm and show convergence for specific variants of MFC focusing on discrete state and action spaces. The mean-field extension of the

linear-quadratic problem has been studied by Carmona et al. (2019a); Wang et al. (2021); Carmona et al. (2021) with policy optimisation methods. Subramanian & Mahajan (2019) also proposes a policy-gradient based algorithm that solves Mean-Field Games and Control problems locally. Additionally, Chen et al. (2021); Li et al. (2021); Wang et al. (2020) consider the approximation error between the Mean-Field and the related Multi-Agent Reinforcement Learning problems. These previous model-free methods are either sample inefficient, assume access to a simulator or consider finite state and action spaces. On the contrary, our algorithm is model-based, sample-efficient, works for continuous spaces and learns the policy online by performing exploration on a real system.

**Main contributions.** We design a novel *Model-based Multi-agent Mean-field Upper Confidence* RL ($M^3$– UCRL) algorithm for the centralized control problem of a large population of collaborative agents. $M^3$– UCRL is practical, performs exploration on a real system (requires no simulator and efficient in terms of samples), and is compatible with neural network policy learning. Our main contributions are the general theoretical regret bounds obtained via a novel mean-field-based analysis, that we also specialize to Gaussian Process models. In the main step of our theoretical analysis, we bound the distance between the mean-field distributions under the true and the approximated dynamics via the model's total estimated uncertainty. Finally, we pose the exploration via entropy maximization problem used in a recent MFG survey (Laurière et al., 2022) in the continuous state and action domain and demonstrate the performance of our algorithm in it. Our results show that $M^3$–UCRL is capable of finding close-to-optimal policies within a few episodes, in contrast to model-free algorithms that require at least six orders of magnitude more samples.

## 2 Problem Statement

We consider the episodic Mean-Field Control (MFC) problem with time-horizon $H$, compact state space $\mathcal{S} \subseteq \mathbb{R}^p$ and compact action space $\mathcal{A} \subseteq \mathbb{R}^q$ that are common for all the agents in the system. We index episodes with the variable $t$ and use $T$ to denote the number of episodes passed. In standard $N$-agent settings, the system's state in episode $t$ at time $h \in \{0, \dots, H-1\}$ is described with individual agents' states $(s_{t,h}^{(1)}, \dots, s_{t,h}^{(N)}) \in \mathcal{S}^N$ which grows exponentially in the number of agents $N$ and represents a common issue in scaling MARL algorithms. In MFCs, we assume that all agents are *identical* and the population is *asymptotically infinite*, i.e., $N \to \infty$, therefore, the agents' states can be described by the *mean-field distribution*:

$$\mu_{t,h}(s) = \lim_{N \to \infty} \frac{1}{N} \sum_{i=1}^{N} \mathbb{I}_{\{s_{t,h}^{(i)}=s\}},$$

where $\mu_{t,h}$ belongs to the space of probability measures $\mathcal{P}(\mathcal{S})$ over $\mathcal{S}$. Due to the assumption of *identical* agents in MFC, we can focus on a *representative agent* from the population that interacts with the distribution of agents instead of individual agents and interactions.

**System Dynamics.** Before every episode $t$, the representative agent selects a policy profile $\boldsymbol{\pi}_t = (\pi_{t,0}, \dots, \pi_{t,H-1})$ where the $H$ individual policies $\pi_{t,h} : \mathcal{S} \times \mathcal{P}(\mathcal{S}) \to \mathcal{A}$ are chosen from a set of admissible policies $\Pi$ (e.g., we parametrize our polices via neural networks; see below). At every time $h$, the representative agent selects an action $a_{t,h} = \pi_{t,h}(s_{t,h}, \mu_{t,h})$. The environment then returns reward $r(s_{t,h}, a_{t,h}, \mu_{t,h})$, and the agent observes its new state $s_{t,h+1}$ and the mean-field distribution $\mu_{t,h+1}$.

In MFCs, the system's dynamics are typically given by a McKean-Vlasov type of stochastic processes and depend on the agent's state, action and the mean-field distribution:

$$s_{t,h+1} = f(s_{t,h}, a_{t,h}, \mu_{t,h}) + \omega_{t,h}, \tag{1}$$

where $\omega_{t,h}$ is an *i.i.d.* additive noise vector. This is in contrast to the standard single-agent model-based RL (cf. Chowdhury & Gopalan (2019); Curi et al. (2020); Kakade et al. (2020)), where the dynamics only depend on agent's action and state. Crucially, since the focus of this work is on *learning* in MFC, we assume that the true dynamics are *unknown*, and the goal is to explore the space and learn about $f$ over a number of episodes. [1] To do so, the representative agent relies on the collected random observations,

---

[1] We expect our results to easily extend and account for unknown rewards by using the same modeling assumptions for the reward function (e.g., as in Chowdhury & Gopalan (2019)).

$\mathcal{D}_t = \{((s_{t,h}, a_{t,h}, \mu_{t,h}), s_{t,h+1})\}_{h=0}^{H-1}$, which come from the interaction with the true system (i.e., from policy rollouts) during episode $t$.

Finally, after every episode, we assume that the whole system is reset and the agent's initial state $s_0$ is drawn from a known initial distribution $\mu_0$, i.e., $s_0 \sim \mu_0$, that remains the same in every episode.

**Mean-field Flow.** Since, in MFC, all agents are identical, interact in a common environment, and follow the *same policy* $\boldsymbol{\pi}_t = (\pi_{t,0}, \ldots, \pi_{t,H-1})$, the subsequent mean-field distributions satisfy the following mean-field flow property as shown by Gu et al. (2020):

**Lemma 1.** *For a given initial distribution $\mu_0$, dynamics $f$ and policy $\boldsymbol{\pi} = (\pi_0, \ldots, \pi_{h-1})$, the mean-field distribution trajectory $\{\mu_h\}_{h=0}^{H-1}$, follows*

$$\mu_{h+1}(ds') = \int_{s \in \mathcal{S}} \mu_h(ds) \mathbb{P}[s_{h+1} \in ds'], \tag{2}$$

*where $\mu_h \in \mathcal{P}(\mathcal{S})$ for all $h \geq 0$, $a_h = \pi_h(s_h, \mu_h)$, $s_{h+1}$ is given by Eq. (1), and $\mu_h(ds) = \mathbb{P}[s_h \in ds]$ under $\boldsymbol{\pi}$.*

We provide the proof in Appendix B.2. To shorten the notation, we use $\Phi(\mu_{t,h}, \pi_{t,h}, f)$ to denote the mean-field transition function from Eq. (2), i.e., we have $\mu_{t,h+1} = \Phi(\mu_{t,h}, \pi_{t,h}, f)$. The lemma shows that the next mean-field distribution explicitly depends on the unknown $f$ (since $s_{t,h+1} = f(s_{t,h}, a_{t,h}, \mu_{t,h}) + \omega_{t,h}$), policy played by the agents $\pi_{t,h}$, and previous state distribution $\mu_{t,h}$.

**Performance metric.** Given a policy profile $\boldsymbol{\pi} = (\pi_1, \ldots, \pi_{H-1})$, the performance of the representative agent is measured via the expected cumulative reward:

$$\mathbb{J}(\boldsymbol{\pi}) = \mathbb{E}\left[ \sum_{h=0}^{H-1} r(s_h, a_h, \mu_h) \right] \tag{3a}$$

$$\text{s.t.} \quad a_h = \pi_h(s_h, \mu_h), \tag{3b}$$

$$s_{h+1} = f(s_h, a_h, \mu_h) + \omega_h, \tag{3c}$$

$$\mu_{h+1} = \Phi(\mu_h, \pi_h, f), \tag{3d}$$

where the expectation is taken over the noise in the transitions and initial distribution $s_0 \sim \mu_0$. By considering the representative agent's perspective, the goal is to discover a *socially* optimal policy $\boldsymbol{\pi}^*$ that maximizes the expected total reward, i.e.,

$$\boldsymbol{\pi}^* \in \arg\max_{\boldsymbol{\pi}} \ \mathbb{J}(\boldsymbol{\pi}) \tag{4}$$

In our theoretical analysis, we make the following assumptions regarding the system's true dynamics, reward function and the set of admissible policies. We use $z, z' \in \mathcal{Z} = \mathcal{S} \times \mathcal{A} \times \mathcal{P}(\mathcal{S})$ to denote $(s, a, \mu)$ and $(s', a', \mu')$, respectively, and we make use of the Wasserstein-1 metric: $W_1(\mu, \mu') = \inf_\nu \int_{\mathcal{S} \times \mathcal{S}} \|x - y\|_2 \, d\nu(x, y)$, where $\nu$ is any probability measure on $\mathcal{S} \times \mathcal{S}$ with marginals $\mu$ and $\mu'$ (see Appendix A for useful properties of $W_1(\mu, \mu')$ that are used in our analysis).

**Assumption 1.** *The transition function $f$ is $L_f$–Lipschitz-continuous, i.e., $\|f(z) - f(z)\|_2 \leq L_f d(z, z')$ where $d(z, z') := \|s - s'\|_2 + \|a - a'\|_2 + W_1(\mu, \mu')$ and $\omega_{t,h}$ are i.i.d. additive $\sigma$-sub-Gaussian noise vectors for all $t \geq 1$ and $h \in \{0, \ldots, H-1\}$.*

**Assumption 2.** *The set of admissible policies $\Pi$ consists of $L_\pi$–Lipschitz-continuous policies such that for any $\pi \in \Pi$: $\|\pi(s, \mu) - \pi(s', \mu')\|_2 \leq L_\pi(\|s - s'\|_2 + W_1(\mu, \mu'))$, and the reward function is $L_r$–Lipschitz-continuous, i.e., $|r(z) - r(z')| \leq L_r d(z, z')$.*

Regularity assumptions like these are standard in the single-agent model-based reinforcement learning literature (Jaksch et al., 2010; Curi et al., 2020; Chowdhury & Gopalan, 2019; Sessa et al., 2022) and mild since, in practice, the policy and reward function classes are typically chosen to satisfy the previous smoothness assumptions. In our experiments (see Section 5), we parametrize our policies with Neural Networks with Lipschitz continuous activations (e.g., tanh, ReLU and linear). We note that the Lipschitzness of such policies then follows from that of the activations when the network's weights are bounded (in practice, this can be done by directly bounding them or via regularization).

**Remark 1.** *The episodic Mean-Field Control problem relates closely to the finite-horizon case of the evolutive Mean-Field Game problem described in Laurière et al. (2022). Both settings consider mean-field distributions evolving over time depending on the agents' policies, however, the MFC problem focuses on collaborative agents while agents in the evolutive MFG are competitive. This distinction leads to different solution concepts of the problems; social welfare in MFC and Nash Equilibrium in MFG.*

# 3 The M³–UCRL Algorithm

In this section, we tackle the Mean-Field Control Problem with unknown dynamics from Eq. (4) by relying on a *model-based* learning scheme. Our *Model-based Multi-agent Mean-field Upper Confidence* RL (M³–UCRL) algorithm uses a statistical model to estimate the system's dynamics and to effectively trade-off between exploration and exploitation during policy learning. Before stating our algorithm, we consider the main properties of the considered dynamics models.

**Statistical Model.** Our representative agent learns about the unknown dynamics from the data collected during the interactions with the environment, i.e., the policy rollouts. In particular, we take a model-based perspective (see Algorithm 1), where the agent models the unknown dynamics and sequentially, after every episode $t$, updates and improves its model estimates based on the previous transition observations, i.e., $\mathcal{D}_1 \cup \cdots \cup \mathcal{D}_t$ where $\mathcal{D}_t = \{((s_{t,h}, a_{t,h}, \mu_{t,h}), s_{t,h+1})\}_{h=0}^{H-1}$ is the set of state transition observations in episode $t$. To reason about plausible models at every episode $t$, the representative agent can take a frequentist or Bayesian perspective. In the first case, it estimates the mean $\boldsymbol{m}_t : \mathcal{Z} \to \mathcal{S}$ and confidence $\boldsymbol{\Sigma}_t : \mathcal{Z} \to \mathbb{R}^{p \times p}$ functions. In the second case, the agent estimates the posterior distribution over dynamical models $p(\tilde{f} | \mathcal{D}_t)$, that leads to $\boldsymbol{m}_t(z) = \mathbb{E}_{\tilde{f} \sim p(\tilde{f} | \mathcal{D}_t)}[\tilde{f}(z)]$, and $\boldsymbol{\Sigma}_t^2(z) = \mathrm{Var}[\tilde{f}(z)]$. We denote $\boldsymbol{\sigma}_t(\cdot) = \mathrm{diag}(\boldsymbol{\Sigma}_t(\cdot))$ and make the following assumptions regarding the considered statistical model irrespective of the taken perspective:

**Assumption 3** (Calibrated model). *The statistical model is calibrated w.r.t. $f$, i.e., there is a known non-decreasing sequence of confidence parameters $\{\beta_t\}_{t \geq 0}$, each $\beta_t \in \mathbb{R}_{>0}$ and depending on $\delta$, such that with probability at least $1 - \delta$, it holds jointly for all $t$ and $z \in \mathcal{Z}$ that $|f(z) - \boldsymbol{m}_t(z)| \leq \beta_t \boldsymbol{\sigma}_t(z)$ elementwise.*

**Assumption 4.** *The function $\boldsymbol{\sigma}_t(\cdot)$ is $L_{\boldsymbol{\sigma}}$-Lipschitz-continuous for all $t \geq 1$.*

The calibrated model assumption (or its equivalents) is standard in online model-based learning (Srinivas et al., 2010; Chowdhury & Gopalan, 2017; 2019; Curi et al., 2020; Sessa et al., 2022). It states that the agent can build high probability confidence bounds around the unknown dynamics at the beginning of every episode. As more observations become available, we expect the *epistemic* uncertainty of the model (that arises due to the lack of data and is encoded in $\boldsymbol{\sigma}(\cdot)$) to shrink, and consequently allow the representative agent to make better decisions as it becomes more confident about the true dynamics. Our theoretical results obtained in Section 4 hold for general model classes as long as Assumption 3 and Assumption 4 are satisfied. Below and in Section 4, we provide concrete conditions (about $f$) and models for which this assumption provably holds.

**Algorithm.** At the beginning of episode $t$, the representative agent constructs the confidence set of dynamics functions, denoted by $\mathcal{F}_{t-1}$, satisfying the elementwise confidence interval in Assumption 3 with $\boldsymbol{m}_{t-1}(\cdot)$ and $\boldsymbol{\sigma}_{t-1}(\cdot)$ estimated based on the observations up until the end of the previous episode $t-1$, i.e.,

$$\mathcal{F}_{t-1} = \left\{ \tilde{f} : |\tilde{f}(z) - \boldsymbol{m}_{t-1}(z)| \leq \beta_{t-1} \boldsymbol{\sigma}_{t-1}(z) \text{ elementwise and } \forall z \in \mathcal{Z} \right\}.$$

Then, the agent selects the optimistic policy $\boldsymbol{\pi}_t$ which achieves the highest possible cumulative reward over the set of admissible policies, $\Pi$, and plausible system dynamics $\mathcal{F}_{t-1}$. In particular, the representative agent solves the following problem:

$$\boldsymbol{\pi}_t = \arg\max_{\boldsymbol{\pi} \in \Pi} \max_{\tilde{f} \in \mathcal{F}_{t-1}} \mathbb{E} \left[ \sum_{h=0}^{H-1} r(\tilde{s}_{t,h}, \tilde{a}_{t,h}, \tilde{\mu}_{t,h}) \right] \tag{5a}$$

$$\text{s.t.} \quad \tilde{a}_{t,h} = \pi_{t,h}(\tilde{s}_{t,h}, \tilde{\mu}_{t,h}), \tag{5b}$$

$$\tilde{s}_{t,h+1} = \tilde{f}(\tilde{s}_{t,h}, \tilde{a}_{t,h}, \tilde{\mu}_{t,h}) + \tilde{\omega}_{t,h}, \tag{5c}$$

$$\tilde{\mu}_{t,h+1} = \Phi(\tilde{\mu}_{t,h}, \pi_{t,h}, \tilde{f}), \tag{5d}$$

---

**Algorithm 1** Model-based RL for Mean-field Control

---

**Input:** Calibrated dynamical model, reward function $r(s_{t,h}, a_{t,h}, \mu_{t,h})$, horizon $H$, initial state $s_{1,0} \sim \mu_0$
  **for** $t = 1, 2, \dots$ **do**
    Use $M^3$–UCRL to select policy profile $\boldsymbol{\pi}_t = (\pi_{t,0}, \dots, \pi_{t,h-1})$ by using the current dynamics model and
    reward function, i.e., solve Eq. (5)
    **for** $h = 0, \dots, H - 1$ **do**
      $a_{t,h} = \pi_{t,h}(s_{t,h}, \mu_{t,h})$,
      $s_{t,h+1} = f(s_{t,h}, a_{t,h}, \mu_{t,h}) + \omega_{t,h}$
      $\mu_{t,h+1} = \Phi(\mu_{t,h}, \pi_{t,h}, f)$
    **end for**
    Update agent's statistical model with new observations $\{(s_{t,h}, a_{t,h}, \mu_{t,h}), s_{t,h+1}\}_{h=0}^{H-1}$
    Reset the system to $\mu_{t+1,0} \leftarrow \mu_0$ and $s_{t+1,0} \sim \mu_{t+1,0}$
  **end for**

---

where the expectation in Eq. (5a) is taken w.r.t. the initial state distribution $\mu_0$ and noise in transitions.

The policy $\boldsymbol{\pi}_t$ found by solving the previous problem is then used in the true multi-agent system (e.g., used by every driver in a ride-hailing service) for one episode. It induces the observed mean-field flow, and aims to improve the total collective reward. After the episode ends, the representative agent augments its observed data, i.e., $\mathcal{D}_{1:t} = \mathcal{D}_{1:t-1} \cup \mathcal{D}_t$, and improves its model estimates $\boldsymbol{m}_t(\cdot)$ and $\boldsymbol{\sigma}_t(\cdot)$. The algorithm is summarized in Algorithm 1. Since solving Eq. (5) is a challenging task even for known dynamics and the focus of this work is on the statistical complexity of learning in MFC (similarly to Kakade et al. (2020); Chowdhury & Gopalan (2019); Curi et al. (2020); Sessa et al. (2022)), we assume that we can solve Eq. (5) for a given $\mathcal{F}_{t-1}$ and $\Pi$. We propose a practical implementation below with details on solving Eq. (5).

The algorithm implements the *upper-confidence bound* principle, since the system's true dynamics $f$ belong to the confidence set $\mathcal{F}_{t-1}$ with high-probability (by Assumption 3). Consequently, the reward achieved by $\boldsymbol{\pi}_t$ under the best possible dynamics $\tilde{f}$ upper bounds the performance of $\boldsymbol{\pi}_t$ under the true dynamics $f$.

**Practical Implementation.** In general, optimizing over the set $\mathcal{F}_{t-1}$ is not tractable. However, a practical problem reformulation has been designed (Moldovan et al., 2015; Curi et al., 2020) for single-agent problems, which defines an auxiliary policy to select the dynamics function $\tilde{f} \in \mathcal{F}_{t-1}$. We generalize this approach to the MFC setting to implement $M^3$–UCRL for our experiments (see Appendix D.1 for details).

$M^3$–UCRL can be combined with any statistical model that satisfies Assumption 3. For example, under mild assumptions on the unknown dynamics, Gaussian Processes (GP) models (Rasmussen, 2003) can be provably calibrated under some regularity assumptions on the dynamics (Srinivas et al., 2010; Abbasi-Yadkori & Szepesvári, 2011). Neural Network (NN) models (Anthony & Bartlett, 2009) such as Probabilistic and Deterministic Ensembles (Lakshminarayanan et al., 2017; Chua et al., 2018) are more scalable, and even though they are not calibrated by default, their empirical recalibration is possible (Kuleshov et al., 2018). Finally, using such differentiable dynamics models and parameterizing $\boldsymbol{\pi}_t(\cdot)$ and the auxiliary policy selecting $\tilde{f}$ via separate neural networks, allows for optimizing Eq. (5) by using gradient-based approaches (see Appendix D).

**Remark 2.** *$M^3$–UCRL uses an optimistic upper-confidence bound strategy similarly to single-agent model-based upper-confidence algorithms (Chowdhury & Gopalan, 2019; Kakade et al., 2020; Curi et al., 2020; Sessa et al., 2022), however, it addresses several important differences: (i) dynamics and rewards explicitly depend on the states of all agents through the mean-field distribution; (ii) each agent's action depends on the mean-field distribution and the evolution of the mean-field trajectory is induced by the collection of actions taken by the agents; (iii) crucially, $M^3$–UCRL employs the principle that each agent in the system executes the same policy selected by the representative agent; Moreover, we note that single-agent MBRL algorithms are inherently unsuitable for solving the Mean Field Control problem.*

## 4 Theoretical Analysis

We analyze our approach using the standard notion of *cumulative regret*. It measures the difference in the cumulative expected reward between the socially optimal policy $\boldsymbol{\pi}^*$ (from Eq. (4)) and the individual

policies selected by M³–UCRL in every episode: $R_T = \sum_{t=1}^{T}(\mathbb{J}(\boldsymbol{\pi}^*) - \mathbb{J}(\boldsymbol{\pi}_t))$. We also use $r_t$ to denote the regret incurred during episode $t$, i.e., $r_t := \mathbb{J}(\boldsymbol{\pi}^*) - \mathbb{J}(\boldsymbol{\pi}_t)$. We aim to show that M³–UCRL achieves *sublinear* cumulative regret, i.e., $\lim_{T\to\infty} \frac{1}{T} R_T = 0$. This implies that as the number of episodes increases, the performance of the selected policies converges to that of the optimal policy: $\mathbb{J}(\boldsymbol{\pi}_t) \to \mathbb{J}(\boldsymbol{\pi}^*)$. We defer all the proofs from this section to Appendix B and only highlight the main contributions of our work.

In general, the speed of convergence of M³–UCRL will depend on the difficulty of estimating $f$. For more complex functions, we expect our models to require more observations to achieve close approximation of the underlying dynamics. Consequently, we use the following model-based complexity measure to quantify this aspect of the learning problem:

$$I_T = \max_{\tilde{D}_1,\dots,\tilde{D}_T} \sum_{t=1}^{T} \sum_{z \in \tilde{D}_1 \cup \dots \cup \tilde{D}_t} \|\boldsymbol{\sigma}_{t-1}(z)\|_2^2, \qquad (6)$$

where $\tilde{D}_t$ is a set of possible observations in an episode for each $t = 1, \dots, T$, and $\boldsymbol{\sigma}_{t-1}(\cdot)$ is the confidence estimate of the selected statistical model computed based on $\tilde{D}_1 \cup \dots \cup \tilde{D}_{t-1}$. We note that the analogous notion of model complexity has been recently used in single agent model-based RL (Chowdhury & Gopalan, 2019; Curi et al., 2020) and the multi-agent setting (Sessa et al., 2022). Intuitively, $I_T$ measures the chosen model's total confidence in estimating the dynamics over $T$ episodes in the worst-case, i.e., when the sets of observations $\tilde{D}_1, \dots, \tilde{D}_T$ are the least informative about the unknown dynamics $f$. For statistical models that fail to learn and generalize, $I_T$ may grow linearly with $T$. On the other hand, for models that learn from a "small" number of samples, we expect $\boldsymbol{\sigma}_{t-1}(\cdot)$ to shrink fast.

**General Model-based Regret Bound.** The following theorem bounds the cumulative regret after $T$ episodes in terms of the model complexity measure $I_T$ defined in Eq. (6).

**Theorem 1.** *Under Assumptions 1 to 4, let $\overline{L}_{T-1} = 1 + 2(1 + L_\pi)[L_f + 2\beta_{T-1}L_\sigma]$, and let $\mu_{t,h} \in \mathcal{P}(\mathcal{S})$ for all $t, h > 0$. Then for all $T \geq 1$ and fixed constant $H > 0$, with probability at least $1 - \delta$, the regret of $M^3$–UCRL is at most:*

$$R_T = \mathcal{O}\Big(\beta_T L_r (1 + L_\pi) \overline{L}_{T-1}^{H-1} \sqrt{H^3 T I_T}\Big).$$

The obtained regret bound explicitly depends on constant factors that describe the environment such as the episode horizon $H$ and the Lipschitz constants. The dependence on the size of the problem, i.e., $p$ and $q$, is implicit in $\beta_T$ and $I_T$ and specific to the statistical model used in the algorithm. We provide more explicit dependencies for Gaussian Processes below and in Appendix C.1. We also note that $\beta_T$ is a function of $\delta$ (see Assumption 3). When polices are selected according to Eq. (5) at every episode, our result implies that the obtained performance $\mathbb{J}(\boldsymbol{\pi}_t)$ eventually converges to the optimal one $\mathbb{J}(\boldsymbol{\pi}^*)$ if the joint dependence on the model-dependent quantities $I_T$ and $\beta_T$ scales as $\mathcal{O}(\sqrt{T})$.

In comparison to other model-based algorithms, both M³–UCRL and H-UCRL (Curi et al., 2020) achieve $\mathcal{O}(\sqrt{H^3 T I_T})$ regret while H-MARL (Sessa et al., 2022) achieves $\mathcal{O}(N^{H/2}\sqrt{H^3 T I_T})$. The additional factor of $\mathcal{O}(N^{H/2})$ comes from the fact that H-MARL considers separate agents with individual action spaces while M³–UCRL optimises for a representative agent instead. In terms of tightness of our results, we expect that the dependency on $\mathcal{O}(\sqrt{T I_T})$ in the regret bound is unavoidable. Scarlett et al. (2017) showed that this is a lower bound for the kernelized bandit problem for specific kernels and using a Gaussian Process. This result relates to the special case of MFC when $H = 1$ and using Gaussian Process statistical model to estimate the transition dynamics. The dependency on $H^3$, however, could be improved under more restrictive assumptions.

In our theoretical analysis, we address non-trivial challenges that arise from considering an infinite number of agents. In particular, we first translate the Mean Field Control to a non-standard MDP with the distributional state space $\mathcal{P}(\mathcal{S})$ and action space $\Pi$ of all admissible policies (for details, see Appendix B.2). This enables us to use the upper-confidence bound principle with the assumption of well-calibrated models to show the following bound on the episodic regret:

$$r_t \leq 2L_r(1 + L_\pi) \sum_{h=1}^{H-1} W_1(\mu_{t,h}, \tilde{\mu}_{t,h}),$$

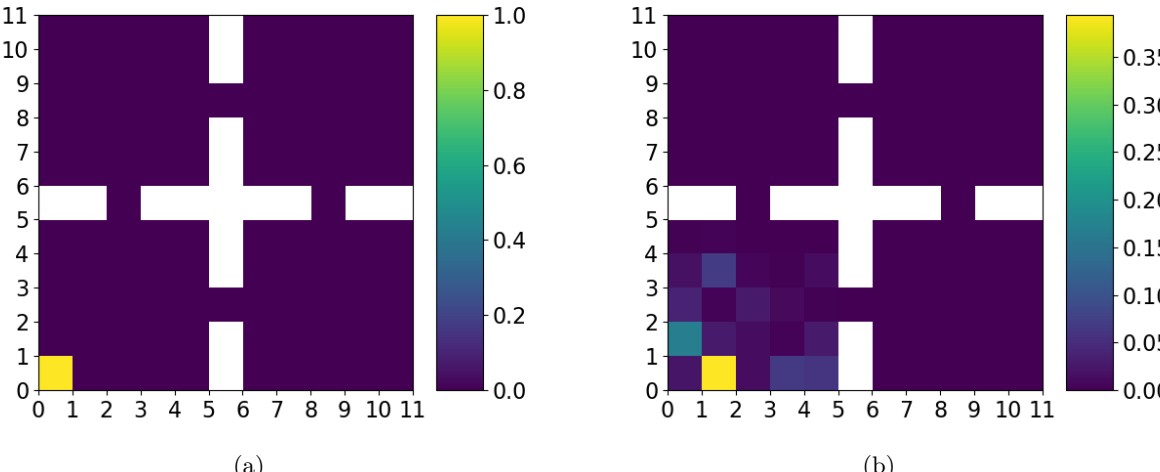

(a)                                         (b)

Figure 1: Fig. 1a shows the initial mean-field distribution $\mu_{h,0}$ for each episode $h$. Fig. 1b shows the mean-field distribution at the end of an episode using a policy that selects actions uniformly at random.

where $\mu_{t,h}$ and $\tilde{\mu}_{t,h}$ are the mean-field distributions when the policy $\boldsymbol{\pi}_t$ is deployed in (i) the true multi-agent system and (ii) the system with dynamics given by the transition function $\tilde{f}_t$ selected by the oracle corresponding to $\boldsymbol{\pi}_t$. The main theoretical contribution of our work is provided in Lemma 5 (see Appendix B.3) which shows the following bound on the Wasserstein-1 distance between the two mean-field distributions:

$$W_1(\tilde{\mu}_{t,h}, \mu_{t,h}) \leq 2\beta_{t-1}\overline{L}_{t-1}^{h-1}\sum_{i=1}^{h-1}\int_{\mathcal{S}}\|\boldsymbol{\sigma}_{t-1}(s, \pi_t(s, \mu_{t,i}), \mu_{t,i})\|_2\,\mu_{t,i}(ds).$$

This shows that the deviation between the used $\tilde{\mu}_{t,h}$ and $\mu_{t,h}$ is bounded by the uncertainty of the statistical model integrated over $\mu_{t,h}$. As the model's uncertainty decreases, the set $\mathcal{F}_t$ shrinks towards the true dynamics $f$ and the dynamics $\tilde{f}$ selected in *Eq.* (5) is restricted to be close to $f$.

Next, we show an example of how the previously obtained regret bounds can be instantiated for particular models. To do so, we consider the case of Gaussian Process models.

**Bounds for Gaussian Process (GP) Models.** GP models are frequently used to model unknown dynamics in the model-based literature (see, e.g., Srinivas et al. (2010); Chowdhury & Gopalan (2019); Curi et al. (2020); Sessa et al. (2022)). They can successfully distinguish between model's and noise uncertainty, while their expressiveness follows from different kernel functions that can be used to model different types of correlation in data.

Under some regularity assumptions on $f$, i.e., when the true $f$ has a bounded norm in the Reproducing Kernel Hilbert Space (RKHS) induced by the GP kernel function, these models provably satisfy Assumption 3. Moreover, we formally show in Appendix C that for such a GP statistical model, we can bound $I_T$ by the *GP maximum mutual information (MMI)*. Similarly, we can express $\beta_T$ via the same quantity, and conclude that sublinearity of our regret depends on the MMI rates. Srinivas et al. (2010) obtain sublinear (in the number of episodes $T$) upper bounds for this quantity in case of the most commonly-used kernels, such as the linear, squared-exponential, or Matérn kernels, while Krause & Ong (2011) prove sublinear bounds for certain composite kernels.

Finally, we note that the aforementioned works consider kernels defined on Euclidean spaces while our input space for $f$, namely $\mathcal{Z} = \mathcal{S} \times \mathcal{A} \times \mathcal{P}(\mathcal{S})$, includes the space of probability densities $\mathcal{P}(\mathcal{S})$. Providing bounds on the GP maximum mutual information capacity in case of kernels defined on probability spaces is an interesting direction for future work.

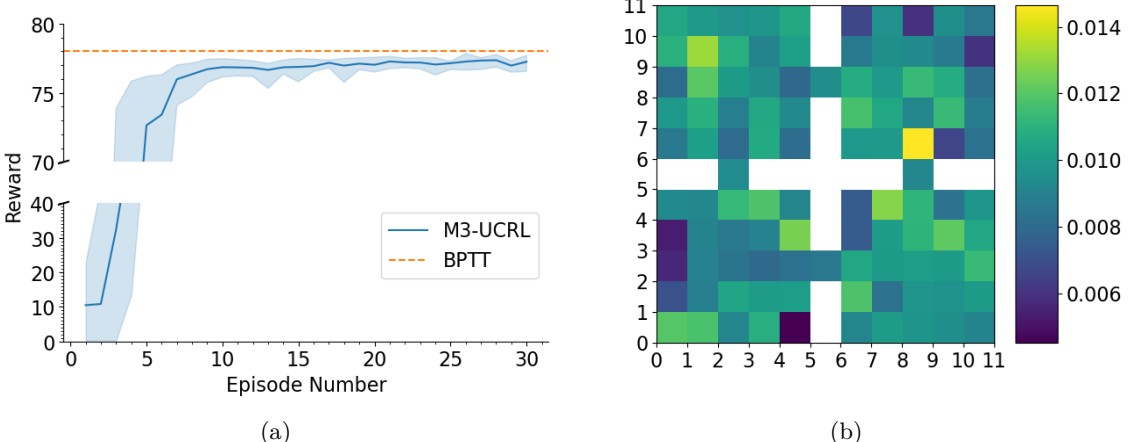

(a)                                                                (b)

Figure 2: Convergence of the $M^3$–UCRL algorithm under system dynamics given by Eq. (7). Fig. 2a shows the rewards defined in Eq. (4) achieved by $M^3$–UCRL over 30 episodes. Confidence bounds show the minimum and maximum over 10 independent experiments. The BPTT line corresponds to the rewards achieved under known dynamics and considered to be optimal in the environment. Fig. 2b shows the close to uniform mean-field distribution at the end of the episode, $h = 20$, for a policy optimised by $M^3$–UCRL.

## 5 Experiments

In this section, we demonstrate the performance of the $M^3$–UCRL algorithm on the *exploration via entropy maximization* problem introduced by Geist et al. (2021) and used as a benchmark problem in a recent survey on Mean-Field Games (Laurière et al., 2022). Different from previous works, we formulate the problem in the continuous state and action spaces as follows. The state-space of the model is described by a 2-dimensional space in $[0, 11]^2$ which is split into 4 equal-sized rooms separated by unit-sized walls with one corridor connecting neighboring rooms (See Fig. 1a). Agents are free to move around within the area and are stopped by the walls if they try to move through them. Each episode $t$ consists of 21 steps starting from $h = 0$ and in each time-step $h$ the representative agent chooses its actions from the action space $\mathcal{A} = [0, 1]^2$. The dynamics of the system in Eq. (1) are of the following form

$$f(s_{t,h}, a_{t,h}, \mu_{t,h}) = s_{t,h} + a_{t,h}, \tag{7}$$

and the additive noise is Gaussian with zero mean, variance $\boldsymbol{\sigma}^2$ and independent dimensions, i.e., $\omega_{t,h} \sim N(0, \boldsymbol{\sigma}^2 I_2)$ for all $t$ and $h$ where $I_2$ is the $2 \times 2$ unit matrix. The individual reward function is defined as $r(s, a, \mu) = -\log(\mu(s))$ meaning that the representative agent aims to avoid crowded areas. On the whole population's level, the average one-step reward is given by $\mathbb{E}_{s \sim \mu}[-\log(\mu(s)] = -\int_{s \in \mathcal{S}} \mu(s) \log(\mu(s))$, so maximizing the average reward is equivalent of maximizing the population's entropy. The population is initially condensed in the bottom-left corner (as shown in Fig. 1a) in a unit square, therefore, a policy is optimal if it disperses the population quickly to reach a uniform distribution.

We parameterize our policy with a Neural Network and use a Deep Ensemble model (Lakshminarayanan et al., 2017) for estimating the system dynamics. To represent the mean-field distribution, we discretize the state-space to unit squares and assign each cell the probability of the representative agent is inside of it. We provide further details on the implementation in Appendix D. Additionally, we ran experiments with Gaussian Process models for the swarm motion problem to showcase the flexibility of design choices in $M^3$–UCRL. Details and results are described in Appendix E.2.

Laurière et al. (2022) shows that exploration is not trivial in the discrete state and action space equivalent of this environment by considering the uniform random policy which fails to reach the uniform distribution within the episode. This result holds in the continuous environment as well as shown by Fig. 1b.

**Results.** Due to the lack of an analytical solution to the problem, we compare the rewards achieved by $M^3$–UCRL in the policy roll-outs (i.e., the interactions with the environment) to the rewards achieved by

the policy optimised via the back-propagation through time (BPTT) algorithm with *known* dynamics. We denote this policy by $\boldsymbol{\pi}^{BPTT}$ and find it to be adequate since when it is used, the population's entropy quickly reaches the maximum achievable corresponding to a uniform distribution. We provide further analysis of this policy in Appendix D.4.

Fig. 2a shows the rewards achieved by $M^3$–UCRL over 30 episodes. The learning can be characterized by two phases; in the first 10 episodes $M^3$–UCRL learns how to navigate the whole state-space, after which it reduces exploration and focuses more on fine-grained improvements. Learning to navigate the environment is challenging due to the precise movements needed to enter the corridors, especially, from an angle. Small changes in the action can decide whether the representative agent moves through the corridor to another room or is stopped by a wall. As shown by the swift improvement in $M^3$–UCRL's performance, it successfully accumulates information in the first phase of the learning process in order to have sufficiently precise policies that can navigate the agent population through the corridors. In the second phase, it gradually refines its policy further to distribute the agents among the rooms quicker and achieve a much closer to uniform distribution as shown on Fig. 2b. The optimised policy after 30 episodes achieves a reward of 77.86 compared to 78.26 that of $\boldsymbol{\pi}^{BPTT}$, i.e., less than 1% difference. [2]

While $M^3$–UCRL converges within 30 episodes, a comparable tabular Q-learning algorithm for the Mean-Field Control problem fails to achieve similar performance in more than 10 million training episodes in the discrete problem defined in Geist et al. (2021) and Laurière et al. (2022). This is most likely due to the large state space, the limitation of deterministic actions, and challenging exploration in the environment. We provide further details of the comparison and why Q-learning requires long training for convergence in Appendix E.1.

## 6  Conclusion

In this work, we propose the first model-based algorithm for the Mean-Field Control problem. The $M^3$–UCRL algorithm runs in episodes, builds optimistic performance estimates, and uses them to efficiently explore the space during policy learning. We proved general regret bounds and showed how they can be specialized to Gaussian Process models. Furthermore, our experiments showcase the practicality of the algorithm and how it can be easily combined with deep Neural Networks. We believe that our approach provides an important step towards making model-based MARL algorithms more principled, scalable, and sample-efficient. We see the following topics important for the further progression of the model-based mean-field learning for both the mean-field control and mean-field game problems: 1) Planning with known dynamics to solve Eq. (5) similarly to Gu et al. (2020); Motte & Pham (2019), 2) Eliminating the need of recalibration to have Neural Networks satisfying Assumption 3, 3) Maximum mutual information bounds for kernels defined on probability spaces.

### Acknowledgments

This project has received funding from the European Research Council (ERC) under the European Unions Horizon 2020 research and innovation program grant agreement No 815943 and ETH Zurich Postdoctoral Fellowship 19-2 FEL-47. This publication was made possible by an ETH AI Center doctoral fellowship to Barna Pasztor.

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

# A    Preliminaries on the Wasserstein Distance

In this section, we provide a brief overview on the Wasserstein distance and its properties used in the proof of our general regret bound in Theorem 1.

**Definition 1** (Wasserstein Distance). *Let $\mu$ and $\nu$ be two probability distributions on $\mathbb{R}^d$, then for $k \in \{1, 2, ...\}$ the Wasserstein distance of order $k$ is defined by*

$$W_k(\mu, \nu) = \left[ \inf_\gamma \int_{\Omega \times \Omega} \|x - y\|^k \, d\gamma(x, y) \right]^{1/k}$$

*where the infimum is taken over all joint distributions $\gamma$ with marginals $\mu, \nu$. For $k = 1$, a standard dual formulation derived by Kantorovich & Rubinstein (1958) is given by:*

$$W_1(\mu, \nu) = \sup_{f \in \mathcal{F}} \left\{ \int f(x) d(\mu - \nu)(x) \right\} \tag{8}$$

*where $\mathcal{F} = \{f : \mathbb{R}^d \to \mathbb{R} \text{ such that } |f(x) - f(y)| \leq \|x - y\| \quad \forall x, y \in \mathbb{R}^d\}$. Both formulations above generalises to laws defined on more general space, e.g., complete and separable metric spaces and infinite-dimensional function spaces such as $L^2[0, 1]$.*

The Wasserstein distance is usually interpreted as the minimum transportation distance between the distributions $\mu$ and $\nu$. For random variables $X \sim \mu$ and $Y \sim \nu$, we use the notations $W_k(X, Y)$ and $W_k(\mu, \nu)$ interchangeably. Assuming that $W_k(X, Y)$ is finite, i.e., $\mathbb{E}[\|X\|^k] + \mathbb{E}[\|Y\|^k] < \infty$, the distance obeys the following properties Panaretos & Zemel (2019):

- For $m \leq n$, by Jensen's Inequality, we have:

$$W_m(X, Y) \leq W_n(X, Y) \tag{9}$$

- For $a \in \mathbb{R}$, it holds

$$W_k(aX, aY) = |a| W_k(X, Y) \tag{10}$$

- For $x \in \mathbb{R}^p$, it holds

$$W_k(X + x, Y + x) = W_k(X, Y) \tag{11}$$

- For $x \in \mathbb{R}^p$, we have that

$$W_2^2(X + x, Y) = \|x + \mathbb{E}[X] - \mathbb{E}[Y]\|^2 + W_2^2(X, Y) \tag{12}$$

**Corollary 1.** *Let $X, Y$ be independent identically distributed random variables on $\mathbb{R}^p$ with $\mathbb{E}[\|X\|^k] < \infty$ and $x, y \in \mathbb{R}^p$. Then,*

$$W_1(X + x, Y + y) \leq \|x - y\|_2$$

*Proof.*

$$
\begin{aligned}
W_2^2(X + x, Y + y) &= W_2^2(X + x - y, Y) && \text{By } Eq.\ (11) \\
&= \|x - y + \mathbb{E}[X] - \mathbb{E}[Y]\|_2^2 + W_2^2(X, Y) && \text{By } Eq.\ (12) \\
&= \|x - y\|_2^2 && X \text{ and } Y \text{ are i.i.d.}
\end{aligned}
$$

Therefore,

$$
\begin{aligned}
W_1(X + x, Y + y) &\leq W_2(X + x, Y + y) && \text{By } Eq.\ (9) \\
&\leq \|x - y\|_2
\end{aligned}
$$

$\square$

# B  Regret Bound Proof

In the following subsections, we present the theoretical analysis that leads to the results of Theorem 1.

## B.1  Measuring performance under arbitrary transition function

Before we start proving the main theorem, we introduce the following notation of $\tilde{\mathbb{J}}(\boldsymbol{\pi}, \tilde{f})$ to measure the performance of a policy $\boldsymbol{\pi}$ under the transition function $\tilde{f}$:

$$\tilde{\mathbb{J}}(\boldsymbol{\pi}, \tilde{f}) := \mathbb{E}\left[\sum_{h=0}^{H-1} r(\tilde{s}_h, \tilde{a}_h, \tilde{\mu}_h)\right] \tag{13a}$$

$$\text{s.t.}\quad \tilde{a}_h = \pi_h(\tilde{s}_h, \tilde{\mu}_h) \tag{13b}$$

$$\tilde{s}_{h+1} = \tilde{f}(\tilde{s}_h, \tilde{a}_h, \tilde{\mu}_h) + \tilde{\omega}_h \tag{13c}$$

$$\tilde{\mu}_{h+1} = \Phi(\tilde{\mu}_h, \pi_h, \tilde{f}), \tag{13d}$$

where the expectation in Eq. (13a) is taken with respect to initial state distribution $\mu_0$ and noise in transitions. We note that we can rewrite the optimization problems in Eq. (4) and Eq. (5) such that $\boldsymbol{\pi}^* \in \arg\max_{\boldsymbol{\pi} \in \Pi} \mathbb{J}(\boldsymbol{\pi}) = \arg\max_{\boldsymbol{\pi} \in \Pi} \tilde{\mathbb{J}}(\boldsymbol{\pi}, f)$ and $\boldsymbol{\pi}_t = \arg\max_{\boldsymbol{\pi} \in \Pi} \max_{\tilde{f} \in \mathcal{F}_{t-1}} \tilde{\mathbb{J}}(\boldsymbol{\pi}, \tilde{f})$, respectively.

## B.2  Measure Formulation

First, we provide the proof of Lemma 1. Then, we reformulate the optimization problems in Eq. (4) and Eq. (5a) as a Markov Decision Process on a probability measure space with transition dynamics defined in Lemma 1. The following proof follows Lemma 2.2 in Gu et al. (2020). We are stating it here for the sake of completeness.

*Proof of Lemma 1.* Fix $\boldsymbol{\pi}$ and let $\varphi$ be a bounded measurable function on $\mathcal{S}$. Recall that $s_h$ denotes random variable of the representative agent's state at time $t$ and, by the definition of $\mu_h$, $s_h \sim \mu_h$.

First note that

$$\mathbb{E}_{s_{h+1}}[\varphi(s_{h+1})] = \int_{s' \in \mathcal{S}} \mu_{h+1}(ds')\varphi(s').$$

On the other hand, by the law of iterated conditional expectation,

$$\begin{aligned}
\mathbb{E}_{s_{h+1}}[\varphi(s_{h+1})] &= \mathbb{E}_{s_0,\dots,s_h}\left[\mathbb{E}_{s_{h+1}}[\varphi(s_{h+1})|s_0,\dots,s_h]\right] \\
&= \mathbb{E}_{s_0,\dots,s_h}\left[\int_{s' \in \mathcal{S}} \varphi(s')\mathbb{P}[s_{h+1} \in ds'|s_0,\dots,s_h]\right] \\
&= \mathbb{E}_{s_0,\dots,s_h}\left[\int_{s' \in \mathcal{S}} \varphi(s')\mathbb{P}[f(s_h, \pi_h(s_h, \mu_h), \mu_h) + \omega_h \in ds']\right] \\
&= \int_{s' \in \mathcal{S}} \varphi(s')\mathbb{E}_{s_0,\dots,s_h}\left[\mathbb{P}[f(s_h, \pi_h(s_h, \mu_h), \mu_h) + \omega_h \in ds']\right] \\
&= \int_{s' \in \mathcal{S}} \varphi(s')\int_{s \in \mathcal{S}} \mu_h(ds)\mathbb{P}[f(s, \pi_h(s, \mu_h), \mu_h) + \omega_h \in ds'].
\end{aligned}$$

In the third equality, we used the dynamics of the system Eq. (1) which states that $s_{h+1}$ is determined by $s_h, a_h, \mu_h$ as $s_{h+1} \sim f(s_h, a_h, \mu_h) + \omega_h$. The last equality follows from $\mu_h(ds) = \mathbb{P}[s_h \in ds]$. Therefore,

$$\int_{s' \in \mathcal{S}} \mu_{h+1}(ds')\varphi(s') = \int_{s' \in \mathcal{S}} \varphi(s')\int_{s \in \mathcal{S}} \mu_h(ds)\mathbb{P}[f(s, \pi_h(s, \mu_h), \mu_h) + \omega_h \in ds'],$$

which implies the desired results

$$\mu_{h+1}(ds') = \int_{s \in \mathcal{S}} \mu_h(ds)\mathbb{P}[f(s, \pi_h(s, \mu_h), \mu_h) + \omega_h \in ds'].$$

$\square$

**Remark 2.** *The relationship described in Lemma 1 is the discrete-time equivalent of the Fokker-Planck (Kolmogorov forward) equation for the McKean-Vlasov Process. While we provided a proof for the dynamics in Eq. (1), it holds for any other functions $\tilde{f}_t \in \mathcal{F}_t$ for $t = 1, 2, \dots$.*

Now, we consider the objective under the true dynamics, Eq. (4). The following lemma is based on Lemma 2.1. in Gu et al. (2020).

**Lemma 2.** *For any policy $\boldsymbol{\pi} = (\pi_0, \dots, \pi_{H-1})$,*

$$\mathbb{J}(\boldsymbol{\pi}) = \sum_{h=0}^{H-1} \hat{r}(\mu_h, \pi_h),$$

*where $\hat{r}$ is the integrated reward function defined as*

$$\hat{r}(\mu, \pi) = \int_{\mathcal{S}} \mu(ds)r(s, \pi(s, \mu), \mu), \tag{14}$$

*and $\mu_h$ is the mean-field distribution induced by the policy $\boldsymbol{\pi}$ following the dynamics defined in Eq. (2).*

*Proof.*

$$
\begin{aligned}
\mathbb{J}(\boldsymbol{\pi}) &= \mathbb{E}_{s_0 \sim \mu_0, \omega_0, \dots, \omega_{H-1}} \left[ \sum_{h=0}^{H-1} r(s_h, \pi_h(s_h, \mu_h), \mu_h) \right] \\
&= \sum_{h=0}^{H-1} \left[ \mathbb{E}_{s_0 \sim \mu_0, \omega_0, \dots, \omega_{h-1}} r(s_h, \pi_h(s_h, \mu_h), \mu_h) \right] \\
&= \sum_{h=0}^{H-1} \int_{\mathcal{S}} \mathbb{P}[s_h \in ds] r(s, \pi_h(s, \mu_h), \mu_h) \\
&= \sum_{h=0}^{H-1} \int_{\mathcal{S}} \mu_h(ds) r(s, \pi_h(s, \mu_h), \mu_h) \\
&= \sum_{h=0}^{H-1} \hat{r}(\mu_h, \pi_h).
\end{aligned}
$$

The expectation in the first line is taken over the random variables $s_0, \omega_0, \dots, \omega_{H-1}$ that induce the state trajectory $s_0, s_1, \dots, s_H$. In particular, for a fixed policy the continuous random variable $s_h$ is a function of $s_0, \omega_0, \dots, \omega_{h-1}$. In the second line, we use the linearity of expectation and $s_h$'s independence of $\omega_h, \dots, \omega_{H-1}$. The third line follows from the definition of expectation.

Note that the fixed and known initial distribution $\mu_0$, the true transition function $f$, and the fixed policy $\boldsymbol{\pi}$ define the mean-field distributions $\mu_h$, for all $0 \le h \le H$ via Eq. (2). The fourth equality follows from Lemma 1, i.e., $\mu_h(ds) = \mathbb{P}[s_h \in ds]$. $\square$

**Corollary 2.** *The arguments in Lemma 2 apply in the case when the transition function of the system is unknown but approximated via $\tilde{f}$. We can rewrite $\tilde{\mathbb{J}}(\boldsymbol{\pi}, \tilde{f})$ defined in Eq. (13a) as*

$$\tilde{\mathbb{J}}(\boldsymbol{\pi}, \tilde{f}) = \sum_{h=0}^{H-1} \hat{r}(\tilde{\mu}_{t,h}, \pi_h).$$

*where $\tilde{\mu}_{t,h}$ is the mean-field distribution induced by the policy $\boldsymbol{\pi}$ under the estimated transition function $\tilde{f}$.*

**Corollary 3.** *By using the objective formulation from Lemma 2, we can restate the optimization problem Eq. (4) as*

$$\max_{\boldsymbol{\pi}} \mathbb{J}(\boldsymbol{\pi}) = \sum_{h=0}^{H-1} \hat{r}(\mu_h, \pi_h) \tag{15}$$

$$s.t. \ \mu_{h+1} = \Phi(\mu_h, \pi_h, f), \tag{16}$$

*where $\Phi$ is defined in Eq. (2). The same holds for the optimization under any other transition function $\tilde{f}$ in Eq. (5).*

**Remark 3.** *Corollary 3 turns the optimization problem Eq. (4) into a Markov Decision Process where the state space is the space of probability measures $\mathcal{P}(\mathcal{S})$ and the action space is the space of functions $\Pi = \{\pi : \mathcal{S} \times \mathcal{P}(\mathcal{S}) \to \mathcal{A}\}$. This reformulation comes with a certain trade-off: $\mathcal{P}(\mathcal{S})$ and $\Pi$ are more complex than $\mathcal{S} \subset \mathbb{R}^q$ and $\mathcal{A} \subset \mathbb{R}^p$ but, in contrast to Eq. (4), the transition function in Eq. (16) is deterministic.*

### B.3  Proof of Theorem 1

As described in Section 3, at the beginning of episode $t$, the representative agent optimizes over both the set of admissible policies, $\Pi$, and plausible transition functions, $\mathcal{F}_{t-1}$, satisfying Assumptions 1 to 3. In the following, we denote the transition function corresponding to the optimal policy by $\tilde{f}_{t-1} \in \mathcal{F}_{t-1}$, i.e.,

$$\boldsymbol{\pi}_t, \tilde{f}_{t-1} = \arg\max_{\boldsymbol{\pi}, \tilde{f}} \tilde{\mathbb{J}}(\boldsymbol{\pi}, \tilde{f}).$$

**Lemma 3.** *Conditioning on the event in Assumption 3 holding true in episode $t$, the following holds for episodic regret:*

$$r_t = \mathbb{J}(\boldsymbol{\pi}^*) - \mathbb{J}(\boldsymbol{\pi}_t) \le \tilde{\mathbb{J}}(\boldsymbol{\pi}_t, \tilde{f}_{t-1}) - \mathbb{J}(\boldsymbol{\pi}_t). \tag{17}$$

*Proof.* Recall that $\boldsymbol{\pi}^*$ denotes the socially optimal policy from Eq. (4) and let $\tilde{f}_{t-1}^* = \arg\max_{\tilde{f} \in \mathcal{F}_{t-1}} \tilde{\mathbb{J}}(\boldsymbol{\pi}^*, \tilde{f})$. The well-calibrated property from Assumption 3 implies that the true transition function $f$ is in the set of functions $\mathcal{F}_{t-1}$ meaning that the true dynamics of the system $f$ has been considered in Eq. (5) at the beginning of episode $t$. Then, since we provide additional flexibility to the optimization, we have

$$\tilde{\mathbb{J}}(\boldsymbol{\pi}^*, \tilde{f}_{t-1}^*) \ge \mathbb{J}(\boldsymbol{\pi}^*)$$

Using this inequality we can bound the episodic regret as follows:

$$\begin{aligned}
r_t = \mathbb{J}(\boldsymbol{\pi}^*) - \mathbb{J}(\boldsymbol{\pi}_t) &\le \tilde{\mathbb{J}}(\boldsymbol{\pi}^*, \tilde{f}_{t-1}^*) - \mathbb{J}(\boldsymbol{\pi}_t) \\
&\le \tilde{\mathbb{J}}(\boldsymbol{\pi}_t, \tilde{f}_{t-1}) - \mathbb{J}(\boldsymbol{\pi}_t),
\end{aligned}$$

where the last inequality comes from the fact that M$^3$–UCRL selects the policy $\boldsymbol{\pi}_t$ according to $(\boldsymbol{\pi}_t, \tilde{f}_{t-1}) = \arg\max_{\boldsymbol{\pi}, \tilde{f}} \tilde{\mathbb{J}}(\boldsymbol{\pi}, \tilde{f})$. $\square$

**Lemma 4.** *Under Assumptions 2 to 3 and conditioning on that the event defined in Assumption 3 holds true, it follows for all $t \ge 1$:*

$$r_t \le \tilde{\mathbb{J}}(\boldsymbol{\pi}_t, \tilde{f}_{t-1}) - \mathbb{J}(\boldsymbol{\pi}_t) \le 2L_r(1 + L_\pi) \sum_{h=1}^{H-1} W_1(\tilde{\mu}_{t,h}, \mu_{t,h}), \tag{18}$$

*where $\mu_{t,h}$ is the mean-field distribution observed in the true unknown environment with dynamics $f$ as in Eq. (3) when agents follow $\boldsymbol{\pi}_t$ and $\tilde{\mu}_{t,h}$ is the mean-field distribution under the approximated dynamics $\tilde{f}_{t-1}$ as in Eq. (13a).*

*Proof.* By Lemma 3, we have that

$$r_t \le \tilde{\mathbb{J}}(\boldsymbol{\pi}_t, \tilde{f}_{t-1}) - \mathbb{J}(\boldsymbol{\pi}_t).$$

By using Lemma 2, Corollary 2 and the triangle inequality we get

$$|\tilde{\mathbb{J}}(\boldsymbol{\pi}_t, \tilde{f}_{t-1}) - \mathbb{J}(\boldsymbol{\pi}_t)| = \left| \sum_{h=0}^{H-1} \hat{r}(\tilde{\mu}_{t,h}, \pi_{t,h}) - \sum_{h=0}^{H-1} \hat{r}(\mu_{t,h}, \pi_{t,h}) \right|$$

$$\leq \sum_{h=0}^{H-1} |\hat{r}(\tilde{\mu}_{t,h}, \pi_{t,h}) - \hat{r}(\mu_{t,h}, \pi_{t,h})|.$$

Consider a fixed $h \in \{0, ..., H-1\}$, then

$$|\hat{r}(\tilde{\mu}_{t,h}, \pi_{t,h}) - \hat{r}(\mu_{t,h}, \pi_{t,h})|$$

$$= \left| \int_{\mathcal{S}} r(s, \pi_{t,h}(s, \tilde{\mu}_{t,h}), \tilde{\mu}_{t,h}) \tilde{\mu}_{t,h}(ds) - \int_{\mathcal{S}} r(s, \pi_{t,h}(s, \mu_{t,h}), \mu_{t,h}) \mu_{t,h}(ds) \right| \qquad \text{From } Eq. \text{ (14)}$$

$$= \left| \int_{\mathcal{S}} r(s, \pi_{t,h}(s, \tilde{\mu}_{t,h}), \tilde{\mu}_{t,h}) \tilde{\mu}_{t,h}(ds) - \int_{\mathcal{S}} r(s, \pi_{t,h}(s, \mu_{t,h}), \mu_{t,h}) \tilde{\mu}_{t,h}(ds) \right.$$

$$\left. + \int_{\mathcal{S}} r(s, \pi_{t,h}(s, \mu_{t,h}), \mu_{t,h}) \tilde{\mu}_{t,h}(ds) - \int_{\mathcal{S}} r(s, \pi_{t,h}(s, \mu_{t,h}), \mu_{t,h}) \mu_{t,h}(ds) \right|$$

$$\leq \int_{\mathcal{S}} |r(s, \pi_{t,h}(s, \tilde{\mu}_{t,h}), \tilde{\mu}_{t,h}) - r(s, \pi_{t,h}(s, \mu_{t,h}), \mu_{t,h}))| \tilde{\mu}_{t,h}(ds) \qquad (19)$$

$$+ \left| \int_{\mathcal{S}} r(s, \pi_{t,h}(s, \mu_{t,h}), \mu_{t,h}) (\tilde{\mu}_{t,h}(ds) - \mu_{t,h}(ds)) \right|, \qquad (20)$$

where the last inequality follows from the triangle inequality and the inequality for the absolute value of definite integrals.

We start by considering the integrand of the term in Eq. (19)

$$|r(s, \pi_{t,h}(s, \tilde{\mu}_{t,h}), \tilde{\mu}_{t,h}) - r(s, \pi_{t,h}(s, \mu_{t,h}), \mu_{t,h})|$$

$$\leq L_r(\|s - s\|_2 + \|\pi_{t,h}(s, \tilde{\mu}_{t,h}) - \pi_{t,h}(s, \mu_{t,h})\|_2 + W_1(\tilde{\mu}_{t,h}, \mu_{t,h}))$$

$$\leq L_r(L_\pi(\|s - s\|_2 + W_1(\tilde{\mu}_{t,h}, \mu_{t,h})) + W_1(\tilde{\mu}_{t,h}, \mu_{t,h}))$$

$$\leq L_r(1 + L_\pi) W_1(\tilde{\mu}_{t,h}, \mu_{t,h}), \qquad (21)$$

where the inequalities follow from Assumption 2.

Next, we consider the function $g(s) = \frac{1}{L_r(1+L_\pi)} r(s, \pi_{t,h}(s, \mu_{t,h}), \mu_{t,h})$

$$|g(x) - g(y)| = \frac{1}{L_r(1 + L_\pi)} |r(x, \pi_{t,h}(x, \mu_{t,h}), \mu_{t,h}) - r(y, \pi_{t,h}(y, \mu_{t,h}), \mu_{t,h})|$$

$$\leq \frac{L_r}{L_r(1 + L_\pi)} (\|x - y\|_2 + \|\pi_{t,h}(x, \mu_{t,h}) - \pi_{t,h}(y, \mu_{t,h})\|_2 + \underbrace{W_1(\mu_{t,h}, \mu_{t,h})}_{=0})$$

$$\leq \frac{L_r}{L_r(1 + L_\pi)} (\|x - y\|_2 + L_\pi(\|x - y\|_2 + \underbrace{W_1(\mu_{t,h}, \mu_{t,h})}_{=0}))$$

$$= \|x - y\|_2,$$

where Assumption 2 has been applied in the second and third lines. Therefore, $g$ is a Lipschitz-1 function.

We bound the second term in Eq. (20) by using the previously defined function $g$

$$\int_{\mathcal{S}} r(s, \pi_{t,h}(s, \mu_{t,h}), \mu_{t,h})(\tilde{\mu}_{t,h}(ds) - \mu_{t,h}(ds))$$

$$= L_r(1 + L_\pi) \int_{\mathcal{S}} \underbrace{\frac{1}{L_r(1 + L_\pi)} r(s, \pi_{t,h}(s, \mu_{t,h}), \mu_{t,h})}_{=g(s)}(\tilde{\mu}_{t,h}(ds) - \mu_{t,h}(ds))$$

$$\leq L_r(1 + L_\pi)W_1(\tilde{\mu}_{t,h}, \mu_{t,h}). \tag{22}$$

The last inequality comes from the Kantorovich-Rubinstein dual formulation of the Wasserstein-1 distance in Eq. (8). Using the inequalities in Eq. (21) and Eq. (22) for Eq. (19) and Eq. (20), respectively, we get that

$$|\hat{r}(\tilde{\mu}_{t,h}, \pi_{t,h}) - \hat{r}(\mu_{t,h}, \pi_{t,h})| \leq 2L_r(1 + L_\pi)W_1(\tilde{\mu}_{t,h}, \mu_{t,h}).$$

Summing it up from $h = 1$ to $h = H - 1$, we obtain the desired result:

$$|\tilde{\mathbb{J}}(\boldsymbol{\pi}_t, \tilde{f}_{t-1}) - \mathbb{J}(\boldsymbol{\pi}_t)| \leq 2L_r(1 + L_\pi) \sum_{h=1}^{H-1} W_1(\tilde{\mu}_{t,h}, \mu_{t,h}).$$

We note that $h = 0$ is removed from the sum since the initial distribution is fixed, $\tilde{\mu}_{t,0} = \mu_{t,0} = \mu_0$, therefore $W_1(\tilde{\mu}_{t,0}, \mu_{t,0}) = 0$. $\qquad\square$

**Lemma 5.** *Under Assumptions 1 to 4, and assuming that event in Assumption 3 holds true, and for $h \in \{1, ..., H\}$ and $t \geq 1$ and fixed policy $\boldsymbol{\pi}_t$, it holds:*

$$W_1(\tilde{\mu}_{t,h}, \mu_{t,h}) \leq 2\beta_{t-1}\overline{L}_{t-1}^{h-1} \sum_{i=0}^{h-1} \int_{\mathcal{S}} \|\boldsymbol{\sigma}_{t-1}(s, \pi_{t,hh}(s, \mu_{t,i}), \mu_{t,i})\|_2 \, \mu_{t,i}(ds), \tag{23}$$

*where $\overline{L}_{t-1} = 1 + 2(1 + L_\pi)[L_f + 2\beta_{t-1}L_{\boldsymbol{\sigma}}]$.*

*Proof.* To simplify the notation when it comes to both $f$ and $\tilde{f}_{t-1}$, we omit the action variable as it is determined via $\boldsymbol{\pi}, \boldsymbol{\mu}$ and the current state, i.e., $f(s, \mu) := f(s, \pi_t(s, \mu), \mu)$ and similarly for $\tilde{f}_{t-1}$. Furthermore, we fix $t \geq 1$.

First, we establish a relationship between $W_1(\tilde{\mu}_{t,h+1}, \mu_{t,h+1})$ and $W_1(\tilde{\mu}_{t,h}, \mu_{t,h})$, i.e., the change in the Wasserstein distance between the state distributions under the true dynamics and the approximated dynamics $\tilde{f}_{t-1}$. Then, we will use this relationship and the fact that the initial distribution is fixed, i.e., $\mu_{t,0} = \tilde{\mu}_{t,0}$, to bound the distance at time $h$.

First, we rewrite $W_1(\tilde{\mu}_{t,h+1}, \mu_{t,h+1})$ in terms of $\pi_{t,h}, \tilde{\mu}_{t,h}$ and $\mu_{t,h}$, use the triangle inequality for the Wasserstein-1 distance, and use the Kantorovich and Rubinstein formulation from Eq. (8).

$W_1(\tilde{\mu}_{t,h+1}, \mu_{t,h+1})$

$= W_1(\Phi(\pi_{t,h}, \tilde{\mu}_{t,h}, \tilde{f}_{t-1}), \Phi(\pi_{t,h}, \mu_{t,h}, f))$

$\leq W_1(\Phi(\pi_{t,h}, \tilde{\mu}_{t,h}, \tilde{f}_{t-1}), \Phi(\pi_{t,h}, \mu_{t,h}, \tilde{f}_{t-1})) + W_1(\Phi(\pi_{t,h}, \mu_{t,h}, \tilde{f}_{t-1}), \Phi(\pi_{t,h}, \mu_{t,h}, f))$

$$= \sup_{v:Lip(v)\leq 1} \int_{s'\in\mathcal{S}} v(s')\left(\int_{s\in\mathcal{S}} \mathbb{P}[\tilde{f}_{t-1}(s, \tilde{\mu}_{t,h}) + \tilde{\omega}_{t,h} \in ds'](\tilde{\mu}_{t,h}(ds) - \mu_{t,h}(ds))\right) \tag{24}$$

$$+ \sup_{v:Lip(v)\leq 1} \int_{s'\in\mathcal{S}} v(s')\left(\int_{s\in\mathcal{S}} \left(\mathbb{P}[\tilde{f}_{t-1}(s, \tilde{\mu}_{t,h}) + \tilde{\omega}_{t,h} \in ds'] - \mathbb{P}[f(s, \mu_{t,h}) + \omega_{t,h} \in ds']\right)\mu_{t,h}(ds)\right). \tag{25}$$

Next, we consider terms from Eq. (24) and Eq. (25) separately, and obtain upper-bounds for both of them in terms of $W_1(\tilde{\mu}_{t,h}, \mu_{t,h})$ and $\boldsymbol{\sigma}_{t-1}(s, \pi_{t,h}(s, \tilde{\mu}_{t,h}), \tilde{\mu}_{t,h})$.

**Step 1: Bounding Eq. (25)** First, we fix $v$ such that $Lip(v) \leq 1$. Then, consider Eq. (25)

$$
\int_{s' \in \mathcal{S}} v(s') \left( \int_{s \in \mathcal{S}} \left( \mathbb{P}[\tilde{f}_{t-1}(s, \tilde{\mu}_{t,h}) + \tilde{\omega}_{t,h} \in ds'] - \mathbb{P}[f(s, \mu_{t,h}) + \omega_{t,h} \in ds']) \mu_{t,h}(ds) \right) \right.
$$

$$
= \int_{s \in \mathcal{S}} \int_{s' \in \mathcal{S}} v(s') \left[ \mathbb{P}[\tilde{f}_{t-1}(s, \tilde{\mu}_{t,h}) + \tilde{\omega}_{t,h} \in ds'] - \mathbb{P}[f(s, \mu_{t,h}) + \omega_{t,h} \in ds'] \right] \mu_{t,h}(ds) \qquad \text{Fubini's thm.}
$$

$$
\leq \int_{s \in \mathcal{S}} W_1(\tilde{f}_{t-1}(s, \tilde{\mu}_{t,h}) + \tilde{\omega}_{t,h}, f(s, \mu_{t,h}) + \omega_{t,h}) \mu_{t,h}(ds) \qquad \text{Eq. (8)}
$$

$$
\leq \int_{s \in \mathcal{S}} \left\| \tilde{f}_{t-1}(s, \tilde{\mu}_{t,h}) - f(s, \mu_{t,h}) \right\|_2 \mu_{t,h}(ds). \qquad \text{Corollary 1} \quad (26)
$$

Note that the state space is compact hence the first moment of $\tilde{\mu}_{t,h+1}$ and $\mu_{t,h+1}$ exist and $W_1(\tilde{\mu}_{t,h+1}, \mu_{t,h+1}) < \infty$, therefore, the integration above is finite and Fubini's theorem is applicable. Next, we consider the following two claims.

**Claim 5.1.** *Under Assumption 3 and assuming that the event defined in it holds true, we have that*

$$
\left\| \tilde{f}_{t-1}(s, \tilde{\mu}_{t,h}) - f(s, \tilde{\mu}_{t,h}) \right\|_2 \leq 2\beta_{t-1} \left\| \boldsymbol{\sigma}_{t-1}(s, \pi_{t,h}(s, \tilde{\mu}_{t,h}), \tilde{\mu}_{t,h}) \right\|_2.
$$

*Proof.*

$$
\left\| \tilde{f}_{t-1}(s, \tilde{\mu}_{t,h}) - f(s, \tilde{\mu}_{t,h}) \right\|_2
$$

$$
= \left\| \tilde{f}_{t-1}(s, \tilde{\mu}_{t,h}) - \boldsymbol{m}_{t-1}(s, \tilde{\mu}_{t,h}) + \boldsymbol{m}_{t-1}(s, \tilde{\mu}_{t,h}) - f(s, \tilde{\mu}_{t,h}) \right\|_2
$$

$$
\leq \left\| \tilde{f}_{t-1}(s, \tilde{\mu}_{t,h}) - \boldsymbol{m}_{t-1}(s, \tilde{\mu}_{t,h}) \right\|_2 + \left\| \boldsymbol{m}_{t-1}(s, \tilde{\mu}_{t,h}) - f(s, \tilde{\mu}_{t,h}) \right\|_2
$$

$$
= \sqrt{\sum_{i=1}^{p} \left[ \tilde{f}_{t-1}(s, \tilde{\mu}_{t,h}) - \boldsymbol{m}_{t-1}(s, \tilde{\mu}_{t,h}) \right]_i^2} + \sqrt{\sum_{i=1}^{p} \left[ \boldsymbol{m}_{t-1}(s, \tilde{\mu}_{t,h}) - f(s, \tilde{\mu}_{t,h}) \right]_i^2}
$$

$$
\leq 2 \sqrt{\sum_{i=1}^{p} \beta_{t-1}^2 \left[ \boldsymbol{\sigma}_{t-1}(s, \pi_{t,h}(s, \tilde{\mu}_{t,h}), \tilde{\mu}_{t,h}) \right]_i^2}
$$

$$
= 2\beta_{t-1} \left\| \boldsymbol{\sigma}_{t-1}(s, \pi_{t,h}(s, \tilde{\mu}_{t,h}), \tilde{\mu}_{t,h}) \right\|_2
$$

where $[\cdot]_i$ denotes the $i$-th coordinate. The first inequality follows from the triangle inequality, after which, we used the event defined in Assumption 3 and the fact that $\tilde{f}_{t-1}$ is chosen to satisfy the same condition. $\square$

**Claim 5.2.** *Under Assumption 1 and Assumption 2, we have that*

$$
\| f(s, \tilde{\mu}_{t,h}) - f(s, \mu_{t,h}) \|_2 \leq L_f(1 + L_\pi) W_1(\tilde{\mu}_{t,h} \mu_{t,h})).
$$

*Proof.* From Assumption 1 and Assumption 2, it follows that

$$
\| f(s, \tilde{\mu}_{t,h}) - f(s, \mu_{t,h}) \|_2 \leq L_f(\| \pi_{t,h}(s, \tilde{\mu}_{t,h}) - \pi_{t,h}(s, \mu_{t,h}) \|_2 + W_1(\tilde{\mu}_{t,h} \mu_{t,h}))
$$

$$
\leq L_f(1 + L_\pi) W_1(\tilde{\mu}_{t,h} \mu_{t,h}).
$$

$\square$

Now, we can combine Claim 5.1 and Claim 5.2 to get

$$
\left\| \tilde{f}_{t-1}(s, \tilde{\mu}_{t,h}) - f(s, \mu_{t,h}) \right\|_2 \tag{27}
$$

$$
\leq \left\| \tilde{f}_{t-1}(s, \tilde{\mu}_{t,h}) - f(s, \tilde{\mu}_{t,h}) \right\|_2 + \| f(s, \tilde{\mu}_{t,h}) - f(s, \mu_{t,h}) \|_2
$$

$$
\leq 2\beta_{t-1} \left\| \boldsymbol{\sigma}_{t-1}(s, \pi_{t,h}(s, \tilde{\mu}_{t,h}), \tilde{\mu}_{t,h}) \right\|_2 + L_f(1 + L_\pi) W_1(\tilde{\mu}_{t,h}, \mu_{t,h}). \tag{28}
$$

Substituting back Eq. (28) into Eq. (26) at the beginning of Step 1, we obtain:

$$\int_{s'\in\mathcal{S}} v(s') \int_{s\in\mathcal{S}} \left[ \mathbb{P}[\tilde{f}_{t-1}(s,\tilde{\mu}_{t,h}) + \tilde{\omega}_{t,h} \in ds'] - \mathbb{P}[f(s,\mu_{t,h}) + \omega_{t,h} \in ds'] \right] \mu_{t,h}(ds)$$

$$\leq \int_{s'\in\mathcal{S}} \left\| \tilde{f}_{t-1}(s,\tilde{\mu}_{t,h}) - f(s,\mu_{t,h}) \right\|_2 \mu_{t,h}(ds)$$

$$\leq L_f(1+L_\pi)W_1(\tilde{\mu}_{t,h},\mu_{t,h}) + 2\beta_{t-1} \int_{s'\in\mathcal{S}} \left\| \boldsymbol{\sigma}_{t-1}(s,\pi_{t,h}(s,\tilde{\mu}_{t,h}),\tilde{\mu}_{t,h}) \right\|_2 \mu_{t,h}(ds). \tag{29}$$

Therefore, we obtain an upper bound for Eq. (25).

**Step 2: Bounding Eq. (24)**

Next, we consider the other part, Eq. (24). Again let $v$ be a fixed function such that $Lip(v) \leq 1$. We have:

$$\int_{s'\in\mathcal{S}} \int_{s\in\mathcal{S}} v(s')\mathbb{P}[\tilde{f}_{t-1}(s,\tilde{\mu}_{t,h}) + \tilde{\omega}_{t,h} \in ds'](\tilde{\mu}_{t,h}(ds) - \mu_{t,h}(ds))$$

$$= \int_{s\in\mathcal{S}} \int_{s'\in\mathcal{S}} v(s')\Big( \mathbb{P}[\tilde{f}_{t-1}(s,\tilde{\mu}_{t,h}) + \tilde{\omega}_{t,h} \in ds'] - \mathbb{P}[f(s,\tilde{\mu}_{t,h}) + \omega_{t,h} \in ds'] \Big)(\tilde{\mu}_{t,h}(ds) - \mu_{t,h}(ds)) \tag{30}$$

$$+ \int_{s\in\mathcal{S}} \int_{s'\in\mathcal{S}} v(s')\mathbb{P}[f(s,\tilde{\mu}_{t,h}) + \omega_{t,h} \in ds'](\tilde{\mu}_{t,h}(ds) - \mu_{t,h}(ds)). \tag{31}$$

We use Fubini's theorem again to change the order of integration, and we consider the inner integration from Eq. (30):

$$\int_{s'\in\mathcal{S}} v(s')\Big( \mathbb{P}[\tilde{f}_{t-1}(s,\tilde{\mu}_{t,h}) + \tilde{\omega}_{t,h} \in ds'] - \mathbb{P}[f(s,\tilde{\mu}_{t,h}) + \omega_{t,h} \in ds'] \Big)$$

$$\leq W_1(\tilde{f}_{t-1}(s,\tilde{\mu}_{t,h}) + \tilde{\omega}_{t,h}, f(s,\tilde{\mu}_{t,h}) + \omega_{t,h}) \qquad\qquad Eq.\ (8)$$

$$\leq \left\| \tilde{f}_{t-1}(s,\tilde{\mu}_{t,h}) - f(s,\tilde{\mu}_{t,h}) \right\|_2 \qquad\qquad Corollary\ 1$$

$$\leq 2\beta_{t-1} \left\| \boldsymbol{\sigma}_{t-1}(s,\pi_{t,h}(s,\tilde{\mu}_{t,h}),\tilde{\mu}_{t,h}) \right\|_2. \qquad\qquad Claim\ 5.1$$

Let $g(s) = \frac{1}{L_{\boldsymbol{\sigma}}(1+L_\pi)} \left\| \boldsymbol{\sigma}_{t-1}(s,\pi_{t,h}(s,\tilde{\mu}_{t,h}),\tilde{\mu}_{t,h}) \right\|_2$, then note that

$$|g(x) - g(y)| = \frac{1}{L_{\boldsymbol{\sigma}}(1+L_\pi)} \left| \left\| \boldsymbol{\sigma}_{t-1}(x,\pi_{t,h}(x,\tilde{\mu}_{t,h}),\tilde{\mu}_{t,h}) \right\|_2 - \left\| \boldsymbol{\sigma}_{t-1}(y,\pi_{t,h}(y,\tilde{\mu}_{t,h}),\tilde{\mu}_{t,h}) \right\|_2 \right|$$

$$\leq \frac{1}{L_{\boldsymbol{\sigma}}(1+L_\pi)} \left\| \boldsymbol{\sigma}_{t-1}(x,\pi_{t,h}(x,\tilde{\mu}_{t,h}),\tilde{\mu}_{t,h}) - \boldsymbol{\sigma}_{t-1}(y,\pi_{t,h}(y,\tilde{\mu}_{t,h}),\tilde{\mu}_{t,h}) \right\|_2$$

$$\leq \frac{1}{1+L_\pi}(\|x-y\|_2 + \|\pi(x,\tilde{\mu}_{t,h}) - \pi(y,\tilde{\mu}_{t,h})\|_2) \qquad\qquad Assumption\ 4$$

$$\leq \|x-y\|_2, \qquad\qquad Assumption\ 2$$

where the first inequality follows from the reverse triangle inequality. Therefore, we can bound Eq. (30) as

$$\int_{s\in\mathcal{S}} \int_{s'\in\mathcal{S}} v(s')\left( \mathbb{P}[\tilde{f}_{t-1}(s,\tilde{\mu}_{t,h}) + \tilde{\omega}_{t,h} \in ds'] - \mathbb{P}[f(s,\mu_{t,h}) + \omega_{t,h} \in ds'] \right)(\tilde{\mu}_{t,h}(ds) - \mu_{t,h}(ds))$$

$$\leq 2\beta_{t-1} \int_{s\in\mathcal{S}} \left\| \boldsymbol{\sigma}_{t-1}(s,\pi_{t,h}(s,\tilde{\mu}_{t,h}),\tilde{\mu}_{t,h}) \right\|_2 (\tilde{\mu}_{t,h}(ds) - \mu_{t,h}(ds))$$

$$= 2\beta_{t-1}L_{\boldsymbol{\sigma}}(1+L_\pi) \int_{s\in\mathcal{S}} g(s)(\tilde{\mu}_{t,h}(ds) - \mu_{t,h}(ds))$$

$$\leq 2\beta_{t-1}L_{\boldsymbol{\sigma}}(1+L_\pi)W_1(\tilde{\mu}_{t,h},\mu_{t,h}). \tag{32}$$

The last inequality comes from the dual formulation of the Wasserstein-1 distance Eq. (8) and the previously shown fact that $g$ is 1-Lipschitz.

Next, we consider the remaining term in Eq. (31), namely,

$$\int_{s\in\mathcal{S}}\int_{s'\in\mathcal{S}} v(s')\mathbb{P}[f(s,\tilde{\mu}_{t,h})+\omega_{t,h}\in ds'](\tilde{\mu}_{t,h}(ds)-\mu_{t,h}(ds)).$$

Let $q(s)=\frac{1}{L_f(1+L_\pi)}\int_{s'\in\mathcal{S}} v(s')\mathbb{P}[f(s,\tilde{\mu}_{t,h})+\omega_{t,h}\in ds']$, then

$$|q(x)-q(y)| = \frac{1}{L_f(1+L_\pi)}\left|\int_{s'\in\mathcal{S}} v(s')\Big(\mathbb{P}[f(x,\tilde{\mu}_{t,h})+\omega_{t,h}\in ds']\right.$$

$$\left. - \mathbb{P}[f(y,\tilde{\mu}_{t,h})+\omega_{t,h}\in ds']\Big)\right|$$

$$\le \frac{1}{L_f(1+L_\pi)}\left|W_1(f(x,\tilde{\mu}_{t,h})+\omega_{t,h}, f(y,\tilde{\mu}_{t,h})+\omega_{t,h})\right| \qquad Eq.\ (8)$$

$$\le \frac{1}{L_f(1+L_\pi)}\|f(x,\tilde{\mu}_{t,h})-f(y,\tilde{\mu}_{t,h})\|_2 \qquad Corollary\ 1$$

$$\le \frac{1}{L_f(1+L_\pi)}\Big(\|x-y\|_2 + \|\pi_{t,h}(x,\tilde{\mu}_{t,h})-\pi_{t,h}(y,\tilde{\mu}_{t,h})\|_2\Big) \qquad Assumption\ 1$$

$$\le \|x-y\|_2. \qquad Assumption\ 2$$

It follows that $q$ is 1-Lipschitz, and therefore

$$\int_{s\in\mathcal{S}}\int_{s'\in\mathcal{S}} v(s')\mathbb{P}[f(s,\mu_{t,h})+\omega_{t,h}\in ds'](\tilde{\mu}_{t,h}(ds)-\mu_{t,h}(ds))$$

$$= L_f(1+L_\pi)\int_{s\in\mathcal{S}} q(s)(\tilde{\mu}_{t,h}(ds)-\mu_{t,h}(ds))$$

$$\le L_f(1+L_\pi)W_1(\tilde{\mu}_{t,h},\mu_{t,h}) \qquad (33)$$

The first equality follows directly from the definition of the function $q$, while the last inequality comes from the Wasserstein Dual formulation of $W_1$ in Eq. (8).

**Combining Step 1 and Step 2:**

By combining Eq. (29), Eq. (32) and Eq. (33), we get

$$W_1(\tilde{\mu}_{t,h+1},\mu_{t,h+1}) \le [2L_f(1+L_\pi)+2\beta_{t-1}L_{\boldsymbol{\sigma}}(1+L_\pi)]W_1(\tilde{\mu}_{t,h}\mu_{t,h}))$$

$$+ 2\beta_{t-1}\int_{s\in\mathcal{S}} \|\boldsymbol{\sigma}_{t-1}(s,\pi_{t,h}(s,\tilde{\mu}_{t,h}),\tilde{\mu}_{t,h})\|_2 \mu_{t,h}(ds).$$

Also, note that

$$\|\boldsymbol{\sigma}_{t-1}(s,\pi_{t,h}(s,\tilde{\mu}_{t,h}),\tilde{\mu}_{t,h})\|_2$$
$$= \|\boldsymbol{\sigma}_{t-1}(s,\pi_{t,h}(s,\tilde{\mu}_{t,h}),\tilde{\mu}_{t,h})-\boldsymbol{\sigma}_{t-1}(s,\pi_{t,h}(s,\mu_{t,h}),\mu_{t,h})+\boldsymbol{\sigma}_{t-1}(s,\pi_{t,h}(s,\mu_{t,h}),\mu_{t,h})\|_2$$
$$\le L_{\boldsymbol{\sigma}}(\|\pi_{t,h}(s,\tilde{\mu}_{t,h})-\pi_{t,h}(s,\mu_{t,h})\|_2 + W_1(\tilde{\mu}_{t,h},\mu_{t,h})) + \|\boldsymbol{\sigma}_{t-1}(s,\pi_{t,h}(s,\mu_{t,h}),\mu_{t,h})\|_2$$
$$\le L_{\boldsymbol{\sigma}}(1+L_\pi)W_1(\tilde{\mu}_{t,h},\mu_{t,h}) + \|\boldsymbol{\sigma}_{t-1}(s,\pi_{t,h}(s,\mu_{t,h}),\mu_{t,h})\|_2,$$

where the first inequality comes from Assumption 4 and the second from Assumption 2. Therefore, we have the following upper-bound on $W_1(\tilde{\mu}_{t,h+1},\mu_{t,h+1})$:

$$W_1(\tilde{\mu}_{t,h+1},\mu_{t,h+1}) \le 2(1+L_\pi)[L_f+2\beta_{t-1}L_{\boldsymbol{\sigma}}]W_1(\tilde{\mu}_{t,h}\mu_{t,h})$$

$$+ 2\beta_{t-1}\int_{s\in\mathcal{S}} \|\boldsymbol{\sigma}_{t-1}(s,\pi_{t,h}(s,\mu_{t,h}),\mu_{t,h})\|_2 \mu_{t,h}(ds).$$

The above inequality establishes the relationship between $W_1(\tilde{\mu}_{t,h+1},\mu_{t,h+1})$ and $W_1(\tilde{\mu}_{t,h},\mu_{t,h})$. We use the fact that $W_1(\tilde{\mu}_{t,0},\mu_{t,0})=0$ and apply this relationship repeatedly $h-1$ times to derive the following upper

bound for $W_1(\tilde{\mu}_{t,h}, \mu_{t,h})$:

$$
\begin{aligned}
&W_1(\tilde{\mu}_{t,h}, \mu_{t,h}) \\
&\leq 2\beta_{t-1} \sum_{i=0}^{h-1} \left[2(1+L_\pi)\left[L_f + 2\beta_{t-1}L_\sigma\right]\right]^{h-1-i} \int_{s\in\mathcal{S}} \|\sigma_{t-1}(s, \pi_{t,i}(s, \mu_{t,i}), \mu_{t,i})\|_2 \, \mu_{t,i}(ds) \\
&\leq 2\beta_{t-1} \sum_{i=0}^{h-1} \Big[\underbrace{1 + 2(1+L_\pi)\left[L_f + 2\beta_{t-1}L_\sigma\right]}_{=:\overline{L}_{t-1}}\Big]^{h-1-i} \int_{s\in\mathcal{S}} \|\sigma_{t-1}(s, \pi_{t,i}(s, \mu_{t,i}), \mu_{t,i})\|_2 \, \mu_{t,i}(ds) \\
&\leq 2\beta_{t-1}\overline{L}_{t-1}^{h-1} \sum_{i=0}^{h-1} \int_{s\in\mathcal{S}} \|\sigma_{t-1}(s, \pi_{t,i}(s, \mu_{t,i}), \mu_{t,i})\|_2 \, \mu_{t,i}(ds).
\end{aligned}
$$

$\square$

**Remark 4.** *Consider the setting of Lemma 5. Note that $\{\mu_{t,i}\}_{i=0}^{h-1}$ is the trajectory of state distributions i.e. $\mathbb{P}[s_{t,i} \in A] = \mu_{t,i}(A)$ for all $A \subseteq \mathcal{S}$ as shown in Lemma 1. Therefore,*

$$
\sum_{i=0}^{h-1} \int_{s\in\mathcal{S}} \|\sigma_{t-1}(s, \pi_{t,i}(s, \mu_{t,i}), \mu_{t,i})\|_2 \, \mu_{t,i}(ds) = \sum_{i=0}^{h-1} \mathbb{E}\left[\|\sigma_{t-1}(z_{t,i})\|_2\right]
$$

$$
= \mathbb{E}\left[\sum_{i=0}^{h-1} \|\sigma_{t-1}(z_{t,i})\|_2\right],
$$

*where $z_{t,i} = (s_{t,i}, \pi_{t,i}(s_{t,i}, \mu_{t,i}), \mu_{t,i})$ and the expectation is taken over the initial distribution and the transition stochasticity. The first equality comes from the aforementioned fact, $\mathbb{P}[s_{t,i} \in A] = \mu_{t,i}(A)$, while the second equality comes from the linearity of expectation.*

*Therefore,*

$$
W_1(\tilde{\mu}_{t,h}, \mu_{t,h}) \leq 2\beta_{t-1}\overline{L}_{t-1}^{h-1}\mathbb{E}\left[\sum_{i=0}^{h-1} \|\sigma_{t-1}(z_{t,i})\|_2\right]. \tag{34}
$$

**Lemma 6.** *Under the setting of Lemma 5 and assuming that the event in Assumption 3 holds true,*

$$
r_t \leq 4\beta_{t-1}L_r(1+L_\pi)\overline{L}_{t-1}^{H-1} H \mathbb{E}\left[\sum_{h=0}^{H-1} \|\sigma_{t-1}(z_{t,h})\|_2\right].
$$

*Proof.* First, note that under the setting of Lemma 6, Lemmas 3 to 5 also hold.

$$
\begin{aligned}
r_t &\leq |\tilde{\mathbb{J}}(\boldsymbol{\pi}_t, \tilde{f}_{t-1}) - \mathbb{J}(\boldsymbol{\pi}_t)| && \textit{Eq. (17)} \\
&\leq 2L_r(1+L_\pi) \sum_{h=1}^{H-1} W_1(\tilde{\mu}_{t,h}, \mu_{t,h}) && \textit{Eq. (18)} \\
&\leq 4\beta_{t-1}L_r(1+L_\pi) \sum_{h=1}^{H-1} \overline{L}_{t-1}^{h-1}\mathbb{E}\left[\sum_{i=0}^{h-1} \|\sigma_{t-1}(z_{t,i})\|_2\right] && \textit{Eq. (34)} \\
&\leq 4\beta_{t-1}L_r(1+L_\pi)\overline{L}_{t-1}^{H-1} H \mathbb{E}\left[\sum_{i=0}^{H-1} \|\sigma_{t-1}(z_{t,i})\|_2\right].
\end{aligned}
$$

In the last inequality, we used the fact that $\overline{L}_{t-1} \geq 1$ hence $\overline{L}_{t-1}^{H-1} \geq \overline{L}_{t-1}^{h-1}$. Furthermore, $\|\sigma_{t-1}(z_{t,i})\|_2 \geq 0$ for any input combination, hence $\sum_{i=0}^{H-1} \|\sigma_{t-1}(z_{t,i})\|_2 \geq \sum_{i=0}^{h-1} \|\sigma_{t-1}(z_{t,i})\|_2$. $\square$

Now, we are ready to prove Theorem 1.

*Proof of Theorem 1.*

$$
\begin{aligned}
R_T^2 &= \left( \sum_{t=1}^{T} r_t \right)^2 \\
&\leq T \sum_{t=1}^{T} r_t^2 \qquad\qquad\qquad\qquad\qquad\qquad\qquad\qquad \text{Cauchy-Schwartz ineq.} \\
&\leq T \sum_{t=1}^{T} 16 p \beta_{T-1}^2 L_r^2 (1+L_\pi)^2 \overline{L}_{T-1}^{2H-2} H^2 \mathbb{E} \left[ \sum_{i=0}^{H-1} \| \boldsymbol{\sigma}_{t-1}(z_{t,i}) \|_2 \right]^2 \qquad \text{Lemma 6} \\
&\leq 16 p T \beta_{T-1}^2 L_r^2 (1+L_\pi)^2 \overline{L}_{T-1}^{2H-2} H^2 \sum_{t=1}^{T} \mathbb{E} \left[ \sum_{i=0}^{H-1} \| \boldsymbol{\sigma}_{t-1}(z_{t,i}) \|_2 \right]^2 \\
&\leq 16 p T \beta_{T-1}^2 L_r^2 (1+L_\pi)^2 \overline{L}_{T-1}^{2H-2} H^3 \mathbb{E} \left[ \sum_{t=1}^{T} \sum_{i=0}^{H-1} \| \boldsymbol{\sigma}_{t-1}(z_{t,i}) \|_2^2 \right] \qquad \text{Jensen's ineq.} \\
&\leq 16 p T \beta_{T-1}^2 L_r^2 (1+L_\pi)^2 \overline{L}_{T-1}^{2H-2} H^3 I_T.
\end{aligned}
$$

The expectations in the expressions above are taken over the initial state and transition noises. For the second inequality, we used that $\beta_{t-1}$ is non-decreasing in $t$ by Assumption 3 while the last inequality follows from $\sum_{t=1}^{T} \sum_{i=0}^{H-1} \| \boldsymbol{\sigma}_{t-1}(z_{t,i}) \|_2^2 \leq I_T$ for all $z_{t,i} \in \mathcal{Z}$ for $t = 1, \ldots, T$ and $i = 0, \ldots, H-1$. Taking square root of both sides yields the desired result. $\qquad\square$

## C Gaussian Processes

In this section, we specialize our main theorem, Theorem 1, to Gaussian Process models.

### C.1 Gaussian Process

The main step of relating Theorem 1 to the Gaussian Process models is to use a Gaussian Process to approximate the unknown dynamics function $f$. In particular, we select a GP with prior $GP(0, k)$ where $k$ is a positive semi-definite kernel on $\mathcal{Z} \times [p]$ where $[p] = \{1, \ldots, p\}$ and $i \in [p]$ specifies the index of the output dimension. The posterior of this Gaussian Process is calculated under the assumption that the observed noise, $\xi_{i,h} = s_{i,h+1} - f(z_{i,h})$, is drawn independently from $\mathcal{N}(0, \lambda \boldsymbol{I}_p)$ for all $i$ and $h$. Here $\lambda$ is a free-parameter that does not necessarily depend on the distribution of $\omega_{t,h}$ for any $t$ and $h$. Then, at the beginning of episode $t$, the posterior mean and variance functions conditioned on $\mathcal{D}_1 \cup \cdots \cup \mathcal{D}_{t-1}$ obtain a closed-form solution (Rasmussen, 2003):

$$
\boldsymbol{m}_{t-1}(z, j) = \boldsymbol{k}_{t-1}(z, j)^\intercal (\boldsymbol{K}_{t-1} + \lambda \boldsymbol{I}_{t-1})^{-1} \boldsymbol{y}_{t-1}, \tag{35a}
$$

$$
\boldsymbol{\sigma}_{t-1}^2(z, j) = k((z,j), (z,j)) - \boldsymbol{k}_{t-1}(z, j)^\intercal (\boldsymbol{K}_{t-1} + \lambda \boldsymbol{I}_{t-1})^{-1} \boldsymbol{k}_{t-1}(z, j), \tag{35b}
$$

where $\boldsymbol{I}_{t-1} \in \mathbb{R}^{(t-1)Hp \times (t-1)Hp}$ is the identity matrix and

$$
\boldsymbol{y}_{t-1} = \left[ [s_{i,h}]_l \right]_{i,h,l=1}^{t-1,H,p} \in \mathbb{R}^{(t-1)Hp},
$$

$$
\boldsymbol{k}_{t-1}(z) = \left[ k((z,j), (z_{i,h}, l)) \right]_{i,h,l=1}^{t-1,H,p} \in \mathbb{R}^{(t-1)Hp},
$$

$$
\boldsymbol{K}_{t-1} = \left[ k((z_{i,h}, l), (z_{i',h'}, l')) \right]_{i,h,l,i',h',l'=1}^{t-1,H,p,t-1,H,p} \in \mathbb{R}^{(t-1)Hp \times (t-1)Hp},
$$

where $[s_{i,h}]_l$ denotes the $l$-th element of the vector $s_{i,h}$. We use the following notation for the mean and variance vectors $\boldsymbol{m}_{t-1}(z) := [\boldsymbol{m}_{t-1}(z, 1), \ldots, \boldsymbol{m}_{t-1}(z, p)]$ and $\boldsymbol{\sigma}_{t-1}^2(z) := [\boldsymbol{\sigma}_{t-1}^2(z, 1), \ldots, \boldsymbol{\sigma}_{t-1}^2(z, p)]$ for all $z$ in $\mathcal{Z}$. In particular, we choose $\lambda = pH$.

## C.2 Assumptions and Relevant Results

We make the following assumptions on the semi-definite kernel $k$ and the true dynamics $f$ of the system.

**Assumption 5.** *The unknown function $f$ belongs to the Reproducing Kernel Hilbert Space (RKHS) of a positive semi-definite kernel $k : (\mathcal{Z} \times [p]) \times (\mathcal{Z} \times [p]) \to \mathbb{R}$, and has bounded RKHS norm, i.e., $\|f\|_k = \sqrt{\langle f, f \rangle_k} \leq B_f$ where $\langle \cdot, \cdot \rangle_k$ is the inner product of the RKHS and $B_f$ is a fixed known positive constant. Furthermore, we assume that $\omega_{t,h}$ is $\sigma$-sub-Gaussian for all $t$ and $h$.*

This assumption is standard when Gaussian Process models are used (e.g., Srinivas et al. (2010); Chowdhury & Gopalan (2019); Curi et al. (2020)).

**Assumption 6.** *The positive semi-definite kernel $k$ is symmetric, continuously differentiable with bounded derivative, and $k(z, z) \leq 1, \forall z \in \mathcal{Z}$. We define the kernel metric as $d_k(z, z') = \sqrt{k(z, z) + k(z', z') - 2k(z, z')}$ and assume that it is Lipschitz-continuous, i.e., $d_k(z, z') \leq L_d d(z, z')$ where $d(z, z') := \|s - s'\|_2 + \|a - a'\|_2 + W_1(\mu, \mu')$ and $L_d$ is some positive constant.*

We note that the previous assumption is very mild since it holds for the most commonly used kernel functions. Assumption 5 and Assumption 6 also imply Assumption 1 which states that the unknown function $f$ is Lipschitz-continuous, i.e., $\|f(z) - f(z')\|_2 \leq L_f d(z, z')$ where $L_f$ is a positive constant (see Corollary 4.36 in Steinwart & Andreas (2008)).

First, we show that under Assumption 5 the Gaussian Process conditioned on $\mathcal{D}_1 \cup \cdots \cup \mathcal{D}_t$ is calibrated for all $t \geq 1$. Our argument follows from the following lemma:

**Lemma 7** (Concentration of an RKHS member, Lemma 5 in Chowdhury & Gopalan (2019) )**.** *Let $k : \mathcal{X} \times \mathcal{X} \to \mathbb{R}$ be a symmetric, positive semi-definite kernel and $f : \mathcal{X} \to \mathbb{R}$ be a member of the RKHS $\mathcal{H}_k(\mathcal{X})$ of real-valued functions on $\mathcal{X}$ with kernel $k$. Let $\{x_t\}_{t \geq 1}$ and $\{\epsilon_t\}_{t \geq 1}$ is conditionally $R$-sub-Gaussian for a positive constant $R$, i.e.,*

$$\forall t \geq 0, \forall \lambda \in \mathbb{R}, \mathbb{E}\left[e^{\lambda \epsilon_t} | \mathcal{F}_{t-1}\right] \leq \exp\left(\frac{\lambda^2 R^2}{2}\right),$$

*where $\mathcal{F}_{t-1}$ is the $\sigma$-algebra generated by $\{x_s, \epsilon_s\}_{s=1}^{t-1}$ and $x_t$. Let $\{y_t\}_{t \geq 1}$ be a sequence of noisy observations at the query points $\{x_t\}_{t \geq 1}$, where $y_t = f(x_t) + \epsilon_t$. For $\lambda > 0$ and $x \in \mathcal{X}$, let*

$$\mu_{t-1}(x) := k_{t-1}(x)^\intercal (K_{t-1} + \lambda I)^{-1} Y_{t-1},$$
$$\sigma_{t-1}^2(x) := k(x, x) - k_{t-1}(x)^\intercal (K_{t-1} + \lambda I)^{-1} k_{t-1}(x),$$

*where $Y_{t-1} := [y_1, \ldots, y_{t-1}]^\intercal$ denotes the vector of observations at $\{x_1, \ldots, x_{t-1}\}$. Then, for any $0 < \delta \leq 1$, with probability at least $1 - \delta$, uniformly over $t \geq 1, x \in \mathcal{X}$,*

$$|f(x) - \mu_{t-1}(x)| \leq \left( \|f\|_k + \frac{R}{\sqrt{\lambda}} \sqrt{2\left(\log(1/\delta) + \frac{1}{2}\sum_{s=1}^{t-1}\log(1 + \lambda^{-1}\sigma_{t-1}^2(x_s))\right)} \right) \sigma_{t-1}(x).$$

We note that similar results have been shown independently in Durand et al. (2018).

Before showing that the above specified Gaussian Process is calibrated, we define the *maximum mutual information*, $\gamma_t$, to measure the maximum information gain the representative agent could possibly obtain on $f$ by observing any set $A \subset \mathcal{Z} \times [p]$ of size $t$.

**Definition 2** (Maximum Mutual Information)**.** *Let $f : \mathcal{X} \to \mathbb{R}$ be a real-valued function defined on a domain $\mathcal{X}$, and $t$ a positive integer. For each subset $A \subset \mathcal{X}$, let $\boldsymbol{y}_A$ denote a noisy version of $\boldsymbol{f}_A$ with $\mathbb{P}[\boldsymbol{y}_A | \boldsymbol{f}_A]$. The Maximum Mutual Information after $t$ noisy observations is defined as*

$$\gamma_t(f, \mathcal{X}) := \max_{A \subset \mathcal{X} : |A| = t} I(\boldsymbol{f}_A; \boldsymbol{y}_A),$$

*where $I(\boldsymbol{f}_A; \boldsymbol{y}_A)$ denotes the mutual information between $\boldsymbol{f}_A$ and $\boldsymbol{y}_A$.*

Using the Maximum Mutual Information, the following corollary shows that the Gaussian Process satisfies Assumption 3 under Assumption 5 and Assumption 6.

**Corollary 4.** *Under Assumption 5 and Assumption 6, with probability $1 - \delta$ for all $t \geq 0$ and $z \in \mathcal{Z}$*

$$\|f(z) - \boldsymbol{m}_t(z)\|_2 \leq \beta_t \|\boldsymbol{\sigma}_t(z)\|_2, \tag{37}$$

*where*

$$\beta_t = B_f + \frac{\sigma}{\sqrt{\lambda}}\sqrt{2(\log(1/\delta) + \gamma_{ptH}(k, \mathcal{Z} \times [p]))}.$$

*Proof.* This corollary follows from Lemma 10 in Chowdhury & Gopalan (2019). $\square$

Additionally, the GP also satisfies Assumption 4 under Assumption 5 and Assumption 6. Lemma 8 follows from Lemma 12 from Curi et al. (2020)

**Lemma 8.** *For all $z$ and $z'$ in $\mathcal{Z}$ and all $t \geq 0$, we have*

$$\|\boldsymbol{\sigma}_t(z) - \boldsymbol{\sigma}_t(z')\|_2 \leq d_k(z, z')$$

*where $d_k$ is the kernel metric defined in Assumption 6.*

We note that Lemma 8 and Assumption 6 immediately imply that $\boldsymbol{\sigma}_t$ satisfies Assumption 4 with Lipschitz constant $L_d$ for all $t \geq 0$.

## C.3 Regret Bound

Now, we are ready to show the regret bound under the choice of a Gaussian Process model.

First, we recall a partial result in Eq. (39) from Lemma 11 in Chowdhury & Gopalan (2019) stating that under Assumption 5 and Assumption 6

$$\sum_{t=1}^{T} \sum_{z \in \tilde{D}_1 \cup \cdots \cup \tilde{D}_{t-1}} \|\boldsymbol{\sigma}_{t-1}(z)\|_2^2 \leq 2epH\gamma_{pHT}(k, \mathcal{Z} \times [p]) \tag{38}$$

where $\tilde{D}_t$ is an arbitrary observation sets during episode $t$ for $t = 1, 2, 3, \ldots$.

**Theorem 2.** *Let $\overline{L}_{T-1} = 1 + 2(1 + L_\pi)[L_f + 2\beta_{T-1}L_{\boldsymbol{\sigma}}]$, and let $\mu_{t,h} \in \mathcal{P}(\mathcal{S})$ for all $t, h > 0$. Then, under Assumption 2, Assumption 5, and Assumption 6 and using a Gaussian Process as described in Appendix C.1, for all $T \geq 1$ and fixed constant $H > 0$, with probability at least $1 - \delta$, the regret of $M^3$–UCRL is at most:*

$$R_T \leq \mathcal{O}\left(\beta_{T-1}L_r(1 + L_\pi)\overline{L}_{T-1}^{H-1}H^2\sqrt{T\gamma_{pHT}(k, \mathcal{Z} \times [p])}\right). \tag{39}$$

*Proof.* As noted before, Assumption 5 and Assumption 6 imply Assumption 1, Assumption 3, and Assumption 4, hence, by Theorem 1, with probability at least $1 - \delta$

$$R_T = \mathcal{O}\left(\beta_T L_r(1 + L_\pi)\overline{L}_{T-1}^{H-1}\sqrt{H^3 T I_T}\right)$$

$$= \mathcal{O}\left(\beta_T L_r(1 + L_\pi)\overline{L}_{T-1}^{H-1}\sqrt{H^3 T \max_{\tilde{D}_1,\ldots,\tilde{D}_T}\sum_{t=1}^{T}\sum_{z \in \tilde{D}_1 \cup \cdots \cup \tilde{D}_t}\|\boldsymbol{\sigma}_{t-1}(z)\|_2^2}\right),$$

where $\overline{L}_{t-1} = 1 + 2(1 + L_\pi)[L_f + 2\beta_{t-1}L_{\boldsymbol{\sigma}}]$. Then we can substitute in Eq. (38) to reach the desired result.

$\square$

Theorem 2 shows that the sublinearity of our regret depends on the Maximum Mutual Information rates, in particular, for sublinear MMI rate the regret is sublinear. Srinivas et al. (2010) consider the most commonly used kernels, and obtain the following sublinear rates for $D \subset \mathbb{R}^d$: Linear kernels $\gamma_t(f, D) = \mathcal{O}(d \log t)$, Squared Exponential kernels $\gamma_t(f, D) = \mathcal{O}((\log t)^{d+1})$, and Matérn kernels $\gamma_t(f, D) = \mathcal{O}(t^{d(d+1)/(2\alpha+d(d+1))}(\log t))$ with $\alpha > 1$. Furthermore, Krause & Ong (2011) show the sublinear property of specific composite kernels. We note that this line of work provides large freedom for kernel design on Euclidean spaces, however, the state space in our problem includes the mean-field distribution space $\mathcal{P}(\mathcal{S})$, therefore, the results are not readily applicable. Proving similar bounds on probability measure spaces is a promising direction for future works.

## D   Experiment Implementation

### D.1   Hallucinated Control Implementation

In Section 3, we introduced M$^3$–UCRL, an algorithm that optimizes over the set of plausible dynamics $\mathcal{F}_{t-1}$ and admissible policies $\Pi$ (see Eq. (5)). However, as noted in the paper, optimizing over $\mathcal{F}_{t-1}$ is intractable in most of the cases. In this sections, we describe a practical and equivalent reformulation of Eq. (5) that helps parametrizing the problem and enables using gradient based optimization to find $\boldsymbol{\pi}_t$ at every episode $t$. First, we introduce an auxiliary function $\eta : \mathcal{Z} \to [-1, 1]^p$ (as in Moldovan et al. (2015); Curi et al. (2020)) where $p$ is the dimensionality of the state space $\mathcal{S}$ and define the following hypothetical dynamics model:

$$\tilde{f}_{t-1}(z) = \boldsymbol{m}_{t-1}(z) + \beta_{t-1}\boldsymbol{\Sigma}_{t-1}(z)\eta(z). \tag{40}$$

Conditioned on the event from Assumption 3 holding true, $\tilde{f}_{t-1}$ for any $\eta$ is calibrated, i.e., $\tilde{f}_{t-1} \in \mathcal{F}_{t-1}$. Furthermore, by the confidence interval in Assumption 3, every function in $\mathcal{F}_{t-1}$ can be written in the form of Eq. (40), i.e.,

$$\forall \tilde{f}_{t-1} \in \mathcal{F}_{t-1}, \quad \exists \eta : \mathcal{Z} \to [-1, 1]^p \quad \text{such that} \quad \tilde{f}_{t-1}(z) = \boldsymbol{m}_{t-1}(z) + \beta_{t-1}\boldsymbol{\Sigma}_{t-1}(z)\eta(z) \quad \forall z \in \mathcal{Z}.$$

Therefore, we can reformulate Eq. (5) as an optimization over the set of admissible policies $\Pi$ and auxiliary function $\eta : \mathcal{Z} \to [-1, 1]^p$ as

$$\boldsymbol{\pi}_t = \arg\max_{\boldsymbol{\pi} \in \Pi} \max_{\eta : \mathcal{Z} \to [-1,1]^p} \mathbb{E}\left[\sum_{h=0}^{H-1} r(\tilde{s}_{t,h}, \tilde{a}_{t,h}, \tilde{\mu}_{t,h})\right] \tag{41a}$$

$$\text{s.t.} \quad \tilde{a}_{t,h} = \pi_{t,h}(\tilde{s}_{t,h}, \tilde{\mu}_{t,h}), \quad \tilde{z}_{t,h} = (\tilde{z}_{t,h}), \tag{41b}$$

$$\tilde{f}_{t-1}(\tilde{z}_{t,h}) = \boldsymbol{m}_{t-1}(\tilde{z}_{t,h}) + \beta_{t-1}\boldsymbol{\Sigma}_{t-1}(\tilde{z}_{t,h})\eta(\tilde{z}_{t,h}), \tag{41c}$$

$$\tilde{s}_{t,h+1} = \tilde{f}_{t-1}(\tilde{z}_{t,h}) + \tilde{\omega}_{t,h}, \tag{41d}$$

$$\tilde{\mu}_{t,h+1} = \Phi(\tilde{\mu}_{t,h}, \pi_{t,h}, \tilde{f}_{t-1}). \tag{41e}$$

We note that for a fixed policy $\pi$, the auxiliary function $\eta(\cdot)$ can be rewritten as $\eta(s, a, \mu) = \eta(s, \pi(s, \mu), \mu) = \eta(s, \mu)$. In essence, this turns the function $\eta(\cdot)$ into an additional policy that exerts "hallucinated" control over the set of plausible models $\mathcal{F}$ (Curi et al., 2020).[3]

Taking expectation in Eq. (41a) still hinders the practicality of the algorithm, however, we note that Corollary 3 applies to Eq. (41) and the following optimization problem is equivalent,

$$\boldsymbol{\pi}_t = \arg\max_{\boldsymbol{\pi} \in \Pi} \max_{\eta : \mathcal{Z} \to [-1,1]^p} \sum_{h=0}^{H-1} \hat{r}(\tilde{\mu}_h, \pi_{t,h}) \tag{42a}$$

$$\text{s.t.} \quad \tilde{f}_{t-1}(\tilde{z}_{t,h}) = \boldsymbol{m}_{t-1}(\tilde{z}_{t,h}) + \beta_{t-1}\boldsymbol{\Sigma}_{t-1}(\tilde{z}_{t,h})\eta(\tilde{z}_{t,h}), \tag{42b}$$

$$\tilde{\mu}_{t,h+1} = \Phi(\tilde{\mu}_{t,h}, \pi_{t,h}, \tilde{f}_{t-1}) \tag{42c}$$

---

[3]We remark that the introduction of $\eta(\cdot)$ policy is a simple trick used before in Moldovan et al. (2015) and Curi et al. (2020) that is suitable for practical implementation of model-based reinforcement learning algortihms.

where $\Phi$ is defined in Eq. (2) and $\hat{r}$ is defined in Eq. (14) as

$$\hat{r}(\mu, \pi) = \int_{\mathcal{S}} \mu(ds) r(s, \pi(s, \mu), \mu).$$

Reformulating the optimization problem as in Eq. (42) and choosing parametrizable functions, e.g., Neural Networks, for $\boldsymbol{\pi}$, $\eta$, and the statistical estimators $\boldsymbol{m}_{t-1}$ and $\boldsymbol{\Sigma}_{t-1}$ allow using gradient ascent to find an optima for Eq. (42a). Even though running gradient ascent does not guarantee the discovery of global optimum, we found in our experiments (see in Section 5) that it still provides a close approximation. We provide further details on the implementation of the transition function $\Phi$ in Appendix D.2

### D.2 Transition Function Implementation

The main challenges of implementing the mean-field distribution function $\Phi$ from Eq. (2) are representing the mean-field distributions and taking the integral over the state space $\mathcal{S}$. For our experiments, we discretize the state-space $\mathcal{S}$ with 104 unit-sized cells. We denote these cells by $C_{ij}$ for $i, j \in \{0, \ldots, 10\}$ where $C_{ij}$ denotes the cell $[i, i+1) \times [j, j+1)$. Note that cells corresponding to walls (see Fig. 1a), e.g., $C_{5,0}$, $C_{5,1}$, or $C_{55}$, are not part of the state-space. For each episode $t$ and time-step $h$, the $i, j$ entry of $\mu_{t,h}$ represents the probability that the representative agent lies in that cell, i.e., $[\mu_{t,h}]_{i,j} = \mathbb{P}[s_{t,h} \in [i, i+1) \times [j, j+1)]$. Furthermore, we denote the middle of each cell $C_{i,j}$ by $c_{i,j}$, i.e., $c_{i,j} = (i + 0.5, j + 0.5)$.

The transition function $\Phi$ is then implemented as follows, for every $i, j \in \{0, \ldots, 10\}$ (for which $C_{ij}$ does not correspond to a wall), every episode $t = 1, 2, \ldots$, and every time-step $h = 0, \ldots, 19$

$$[\mu_{t,h+1}]_{i,j} = \sum_{k,l} \mathbb{P}[f(c_{k,l}, \pi_{t,h}(c_{k,l}, \mu_{t,h}), \mu_{t,h}) + \omega_{t,h} \in C_{i,j}][\mu_{t,h}]_{k,l}$$

We assume that the noise term $\omega_{t,h}$ is Gaussian with 0 mean and $\boldsymbol{\sigma}^2$ variance and the two dimensions are independent, therefore,

$$\mathbb{P}[f(c_{k,l}, \pi_{t,h}(c_{k,l}, \mu_{t,h}), \mu_{t,h}) + \omega_{t,h} \in C_{i,j}]$$
$$= \mathbb{P}[f(c_{k,l}, \pi_{t,h}(c_{k,l}, \mu_{t,h}), \mu_{t,h})_1 + \omega_{t,h,1} \in [i, i+1)] \times \mathbb{P}[f(c_{k,l}, \pi_{t,h}(c_{k,l}, \mu_{t,h}), \mu_{t,h})_2 + \omega_{t,h,2} \in [j, j+1)]$$
$$= (\phi(i - f(c_{k,l}, \pi_{t,h}(c_{k,l}, \mu_{t,h}), \mu_{t,h})_1) - \phi(i + 1 - f(c_{k,l}, \pi_{t,h}(c_{k,l}, \mu_{t,h}), \mu_{t,h})_1))$$
$$\times (\phi(j - f(c_{k,l}, \pi_{t,h}(c_{k,l}, \mu_{t,h}), \mu_{t,h})_2) - \phi(j + 1 - f(c_{k,l}, \pi_{t,h}(c_{k,l}, \mu_{t,h}), \mu_{t,h})_2))$$

where $\phi$ is the cumulative distribution function of $N(0, \boldsymbol{\sigma}^2)$.

In the implementation, we also redistribute some of the probability mass during transition based on the walls location. If there is a wall between $C_{i,j}$ and $f(c_{k,l}, \pi_{t,h}(c_{k,l}, \mu_{t,h}), \mu_{t,h}) + \omega_{t,h}$, we set $\mathbb{P}[f(c_{k,l}, \pi_{t,h}(c_{k,l}, \mu_{t,h}), \mu_{t,h}) + \omega_{t,h} \in C_{i,j}] = 0$. Similarly, for $C_{i,j}$ adjacent to a wall or border we increase $\mathbb{P}[f(c_{k,l}, \pi_{t,h}(c_{k,l}, \mu_{t,h}), \mu_{t,h}) + \omega_{t,h} \in C_{i,j}]$ by the amount that is set to zero due to the adjacent wall.

### D.3 Transition Model estimation

To estimate the unknown transition function $f$ in Section 5, we use deep ensemble models (Lakshminarayanan et al., 2017). In particular, we use an ensemble of 10 feed-forward Neural Networks (NNs) with one hidden layer of size 32 and leaky ReLu action functions. We use two output layers joined to the hidden middle layer. The first one uses linear activation and returns the mean of the function while the second returns the estimated variance using softplus activation. We follow the optimisation procedure described in (Lakshminarayanan et al., 2017) that minimizes the negative log-likelihood for each NN under the assumption of heteroscedastic Gaussian noise. We included the adversarial training procedure as well for robustness and smoothing. In prediction time, the ensemble estimates the mean and variance of the unknown function via the empirical mean and variance of the Neural Networks' mean outputs.

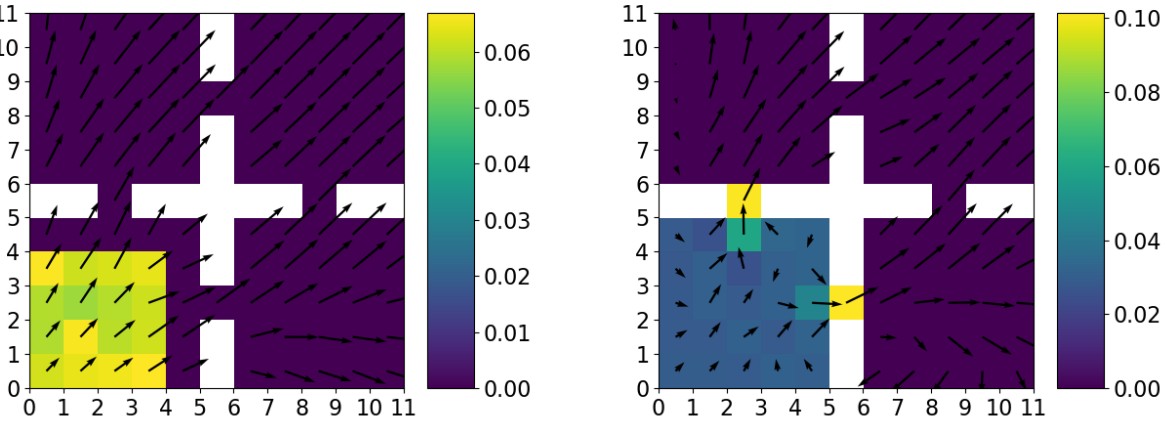

(a) Mean-field distribution under the BPTT optimised policy at time-step $h = 3$.

(b) Mean-field distribution under the BPTT optimised policy at time-step $h = 5$.

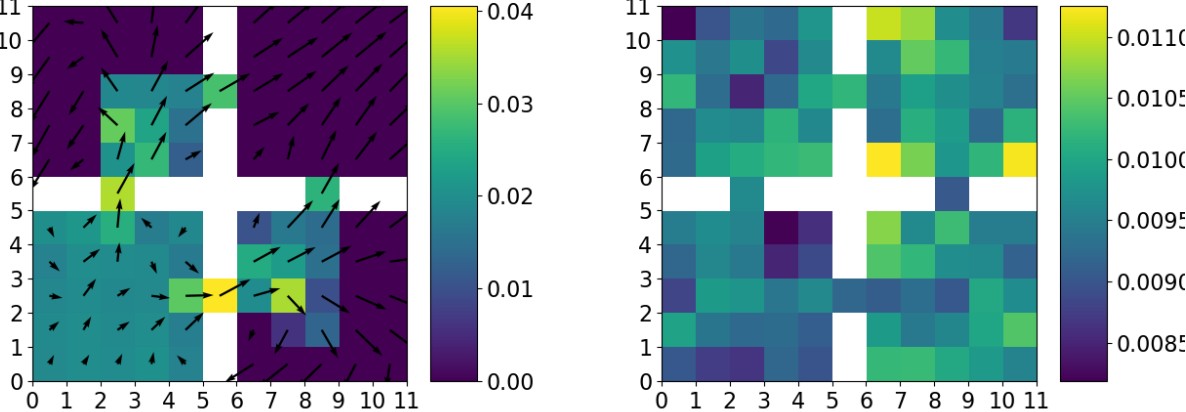

(c) Mean-field distribution under the BPTT optimised policy at time-step $h = 8$.

(d) Mean-field distribution under the BPTT optimised policy at time-step $h = 20$.

Figure 3: Mean-field distribution following the policy optimised via BPTT at four time-steps ($h = 3, 5, 8, 20$) in a single policy roll-out. Arrows depict the actions taken by the agents in the corresponding squares. Note that the coloring range changes with the time-steps.

## D.4 Policy optimisation with known environment

We consider the policy optimised via back-propagation through time, $\pi^{BPTT}$, as a reasonable point-of-reference for our convergence analysis as it successfully disperses the population in the state space quickly without any notable improvements in its strategy.

Fig. 3 shows the mean-field distribution over one episode following $\pi^{BPTT}$ [4]. As shown on Fig. 3a, in the first few steps $\pi^{BPTT}$ spreads the distribution evenly in the bottom-left corner, however, in the next two steps as shown on Fig. 3b it concentrates a larger mass into the corridors to the neighbouring rooms. It is an optimal choice of action because the corridors act as bottlenecks hence the more it can push through at once the easier it will be to achieve a uniform distribution in the latter time-steps. In the next 3 steps until $h = 8$, $\pi^{BPTT}$ further propagates a larger portion of the population mass to the empty top-right room. In the meanwhile, it starts to distribute the population in the top-left and bottom-right rooms as well. In the following time-steps, $\pi^{BPTT}$ focuses on achieving an even distribution within the rooms without pushing

---

[4]The supplementary material includes animation of the whole episode named as `known_dynamics_animation.mp4`

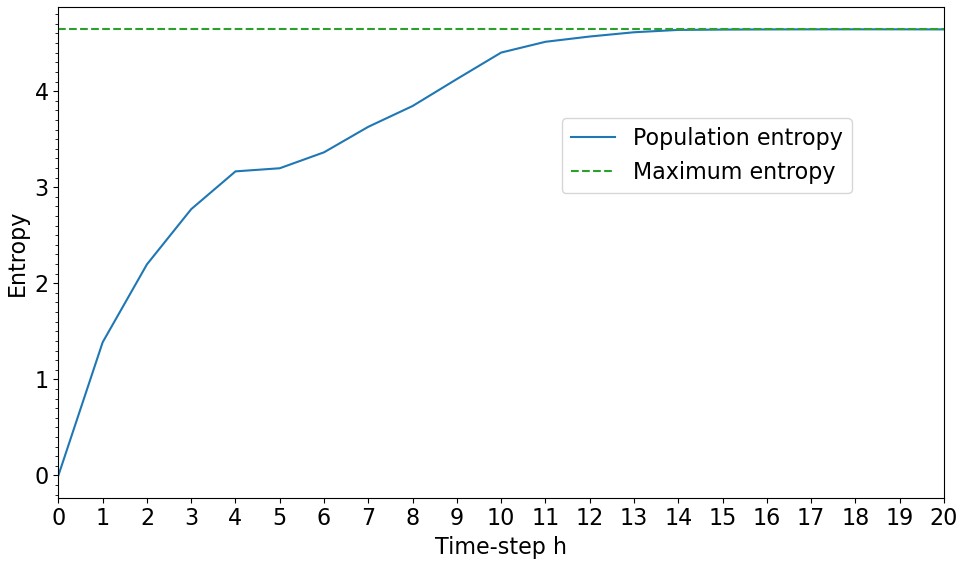

Figure 4: Reward and population action size per time-step in one episode. The policy followed by the agents were optimised via BPTT.

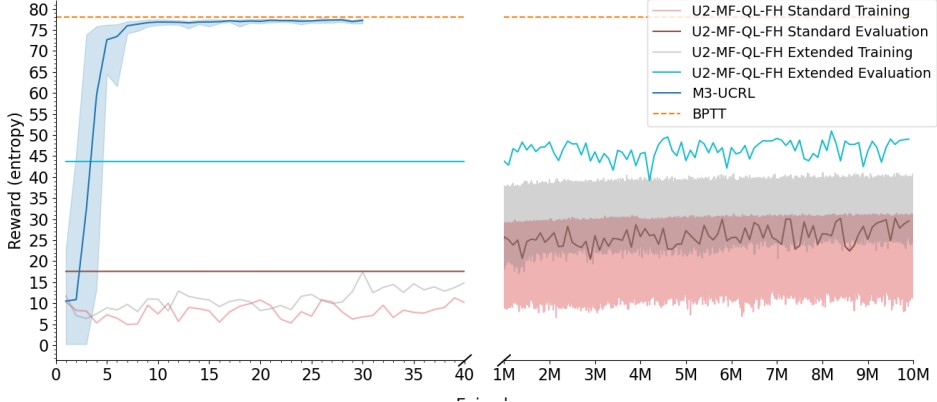

Figure 5: Rewards achieved by *U2-MF-QL-FH* over 10 million training episodes. *Training Rewards* show the estimated reward by *U2-MF-QL-FH* while *Evaluation Rewards* show the performance of the greedy policy in a roll-out. The policy is evaluated after every million training episodes. The hyperparameters of the training were $\omega^Q = 0.7, \omega^\nu = 0.05$ and $\epsilon = 0.15$.

significant population mass through the corridors. Finally, Fig. 3d shows the mean-field distribution at the end of the episode, $h = 20$, when the population is distributed uniformly and maximized its entropy. We provide a further animation with every time-steps as part of the supplementary material.

Fig. 4 shows the mean-field distribution entropy over the time-steps of the same episode as in Fig. 3. $\pi^{BPTT}$ successfully increases the entropy until time-step 14 where it reaches the maximum achievable by the uniform distribution and then maintains this level. The small stagnation in entropy from $h = 4$ to $h = 5$ is due to the bottleneck of the corridors as shown on Fig. 3b.

# E  Additional Experiments

## E.1  Comparison to model-free reinforcement learning

$M^3$–UCRL is a unique algorithm in the Mean-Field Control literature because it considers continuous state and action spaces and finite time-horizon. As described in Section 2, the goal in this setting is to find an optimal policy that considers the evolution of the mean-field distribution over one episode for a fixed initial distribution. In comparison, the majority of the works consider finite spaces and infinite time-horizon in which the goal is to find the optimal stationary pair of policy and mean-field distribution. We note that this is a significantly different solution concept to ours, hence, proposed algorithms are not suitable to solve the exploration via entropy maximization problem described in Section 5. To the best of our knowledge, the only algorithm that considers a finite time-horizon problem is the *U2-MF-QL-FH* proposed by Angiuli et al. (2021). This algorithm is a Q-Learning variant that optimises the Q values via a two timescale approach. Even though, the problem formulation for *U2-MF-QL-FH* is similar to ours described in Section 2, there are notable differences. First, *U2-MF-QL-FH* is proposed for finite state and action spaces, uses a tabular representation for the Q values and select deterministic actions. Second, it is model-free and assumes access to simulator of one-step transitions. On the other hand, $M^3$–UCRL does not rely on a tabular representation, learns in continuous spaces, and have limited interactions with the environment to reduce costs associated with these interactions.

Due to the formulation of *U2-MF-QL-FH*, we ran experiments in the finite state and action space variant of the Exploration via entropy maximization problem as described in Geist et al. (2021); Laurière et al. (2022). As we demonstrate below, the deterministic policy used by *U2-MF-QL-FH* is a significant limitation, therefore, we also alter its action space to be a set of 9 distributions over the 5 potential actions from which *U2-MF-QL-FH* chooses [5]. We call the former *Standard* formulation and the latter *Extended*.

As shown on Fig. 5, *U2-MF-QL-FH* converges slowly with high variance in rewards for both formulations, however, the *Extended* version outperforms the *Standard* version for both training and evaluation performance as expected. One of the main factors for slow learning is that the Q values are hard to estimate due to the large number of state-action-time pairs and infrequent visits of many states. In particular, the Q value table has $|\mathcal{S}| \times |\mathcal{A}| \times H = 104 \times 5 \times 20 = 10,400$ entries ($18,720$ for the *Extended* version), and states far from the bottom left corner are rarely visited because they can be reached only in the later steps of an episode. While extending the action-space to non-deterministic actions clearly improves performance, it increases significantly the tabular representation leading to even longer training required. Even after 10 million episodes, we observed large changes in the Q value table indicating that the algorithm would need further iterations, but we had to terminate it due to computational limits. These results further highlights the contribution of $M^3$–UCRL to the literature.

## E.2  Swarm Motion Experiments

To complement experiments in Section 5, we also implemented $M^3$–UCRL with Gaussian Process models for the *swarm motion* problem. This setting considers an asymptotically large ($N \to \infty$) population of agents moving around in a space, maximizing a location-dependent reward function, and avoiding congested areas (Almulla et al., 2017).

We choose the state space $\mathcal{S}$ as the $[0,1]$ interval with periodic boundary conditions, i.e., the unit torus, and the action space $\mathcal{A}$ as the compact interval $[-7,7]$. Each episode partitions the unit time length into $H = 200$ equal steps of length $\Delta h = 1/200$. The unknown dynamics of the system is given by

$$f(s,a) = s + a\Delta h, \tag{43}$$

and $\omega_{t,h} \sim N(0, \Delta h)$ for all $t$ and $h$. We note that, for now, the dynamics do not depend on the mean-field distribution, but later on in Eq. (45), we also consider such a case. The reward function is of the form

---

[5]The set of distribution consists of the following: 5 deterministic actions (4 directions plus staying idle), 4 $50\% - 50\%$ split between adjacent moves (up-right, right-down, down-left, left-up) and 1 distribution that assigns 25% to all 4 directions. Note that the action space of the *Standard* formulation is a subset of this action space, therefore, it is expected that the *Extended* version achieves better performance.

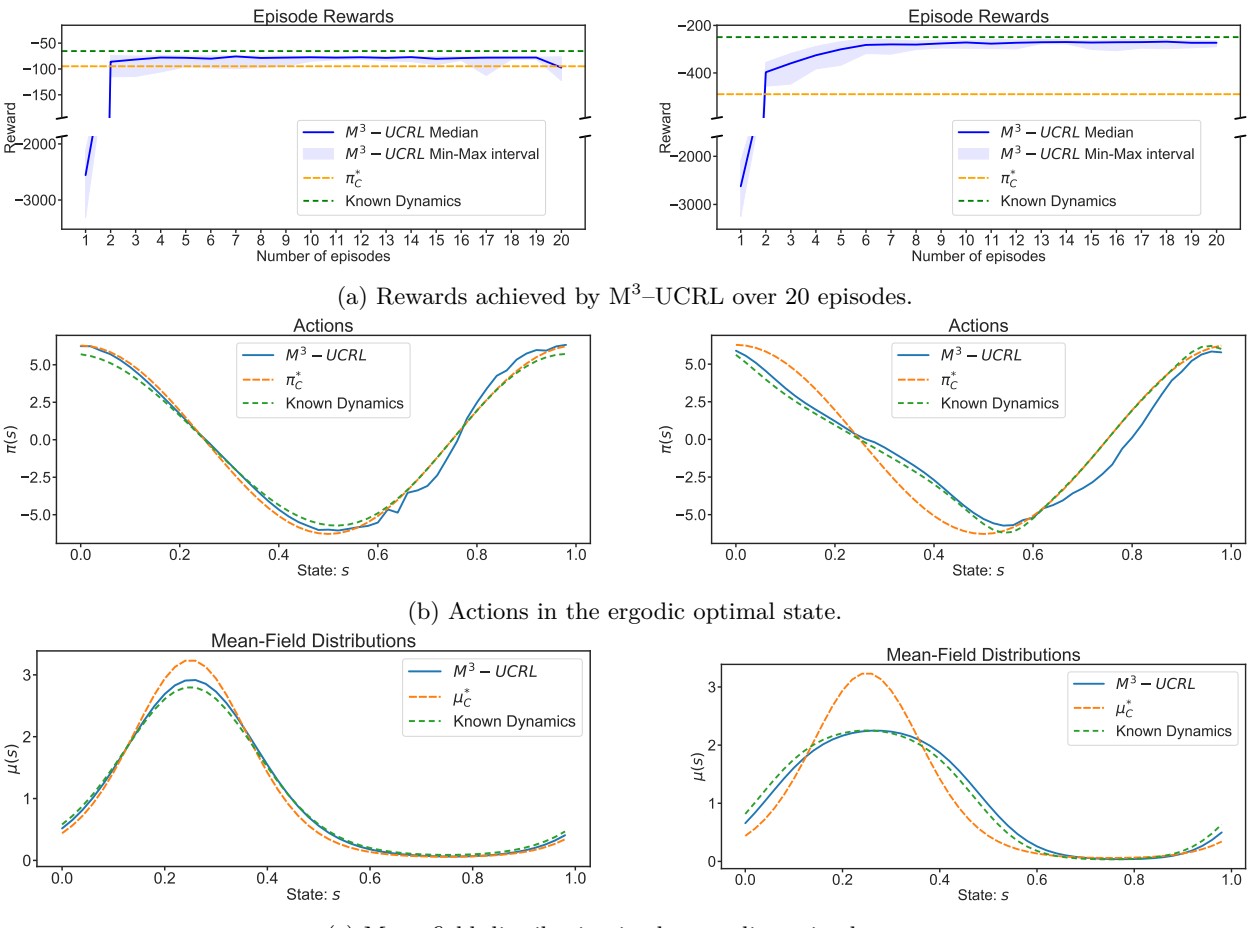

(a) Rewards achieved by $M^3$–UCRL over 20 episodes.

(b) Actions in the ergodic optimal state.

(c) Mean-field distribution in the ergodic optimal state.

Figure 6: Results for dynamics model: **(Left column)** $f(s, a) = s + a\Delta h$ , and **(Right column)** $f(s, a, \mu) = s + a(4 - 4\mu(s))\Delta h$. $(\pi_C^*, \mu_C^*)$ represent an analytic solution to the continuous time swarm motion problem, and thus, it is not discrete-time optimal. The main discrete-time benchmark is Known Dynamics that unlike our $M^3$–UCRL knows the true dynamics. In both cases (left and right), $M^3$–UCRL converges in a small number of episodes and discovers a near-optimal policy.

$r(s, a, \mu) = \varphi(s) - \frac{1}{2}|a| - \ln(\mu(s))$, where the second term penalizes large actions, the third term introduces the aversion to crowded regions and the first term defines the reward received at the position $s$, namely, $\varphi(s) = -2\pi^2[-\sin(2\pi s) + |\cos(2\pi s)|^2] + 2\sin(2\pi s)$.

The advantage of this task is that one can analytically obtain the following optimal solution in the *continuous-time* setting when $\Delta h \to 0$ and the time horizon is infinite, i.e., $H \to \infty$ Almulla et al. (2017):

$$\pi_C^*(s, \mu) = 2\pi \cos(2\pi s),$$
$$\mu_C^*(s) = \frac{\exp(2\sin(2\pi s))}{\int \exp(2\sin(2\pi s'))ds'}, \tag{44}$$

where $\pi_C^*$ and $\mu_C^*$ form an ergodic solution, i.e., $\mu_C^* = \Phi(\mu_C^*, \pi_C^*, f)$. We use the continuous time optimal solutions from Eq. (44) (where the subscript $C$ stands for continuous-time solutions) as a benchmark for comparison, however, we note that our time discretization introduces certain deviations. Consequently, $\pi_C^*$ might no longer be optimal, and so we also compare our solution to the one obtained under the same setup of discrete time steps and optimization procedure when the true dynamics are *known* (Known Dynamics). This corresponds to our main benchmark.

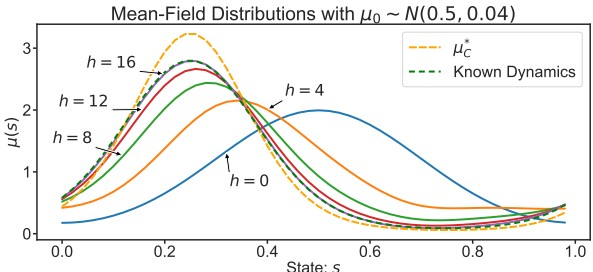 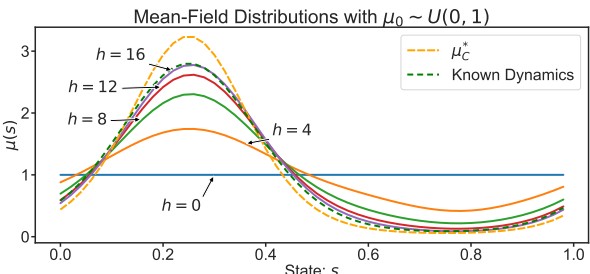

Figure 7: Mean-Field distributions over one episode with different initial distributions and dynamics given by Eq. (43) after training has converged. (**Left side**) $\mu_0 \sim \mathcal{N}(0.5, 0.04)$ (**Right side**) $\mu_0 \sim \mathcal{U}(0, 1)$. Already at $h = 16$, the distribution induced by M$^3$–UCRL policy nearly-matches the one when dynamics are known.

Finally, we also consider the modified swarm motion problem with dynamics given by

$$f(s, a, \mu) = s + a(4 - 4\mu(s))\Delta h. \tag{45}$$

The additional term that depends on the mean-field distribution models the effect of congestion on the transition. Specifically, in comparison to Eq. (43), the more congested an area is, the larger the action an agent must exert to achieve the same movement.

While swarm motion resembles the exploration via entropy maximization problem in Section 5, there are meaningful differences besides the lower dimensional state and action spaces. Namely, certain areas in the state space rewarded differently and large actions are explicitly penalized, therefore, agents not only have to explore the state space but also learn the trade-off between areas and actions. Eq. (45) further extends the swarm motion problem with a dependency on the mean-field distribution that increases the complexity of the agent's learning problem.

Fig. 6 depicts the results when the true dynamics are given by Eq. (43) (Left column) and Eq. (45) (Right column). In Fig. 6a (Left), we show the rewards achieved over 20 episodes. In Fig. 6b (Left) and Fig. 6c (Left), we show the ergodic solution of M$^3$–UCRL and compare it with $(\pi_C^*, \mu_C^*)$ and the solution found when dynamics are known. As expected, the introduced time-discretization results in some deviations from Eq. (44) which allow the policy found when the dynamics are known to exploit them and achieve higher reward. On the other hand, M$^3$–UCRL converges in a small number of episodes and drives the environment into an ergodic state with policy and mean-field distribution almost identical to the solution under known dynamics (see Fig. 6b (Left) and Fig. 6c (Left)) . We note that, while minimal fluctuations in the episodic rewards are present, the M$^3$–UCRL is robust in finding close to optimal solutions.

In Fig. 6 (Right column), we show the results obtained for the dynamics from Eq. (45). Even though $\pi^*$ and $\mu^*$ in Eq. (44) are no longer applicable for this problem, we include them in the figures to show the effect of the introduced congestion term. Similarly to our previous experiments, M$^3$–UCRL successfully converges to the solution obtained under known dynamics. Fig. 6b (Right) shows that the two policies are not completely aligned after the same number of episodes as before, which is due to the harder estimation problem and larger confidence estimates.

In the previous experiments, we selected $\mu^*$ as the initial distribution to focus on finding an optimal policy function $\pi$. To evaluate the robustness of M$^3$–UCRL and its ability to drive an arbitrary initial distribution towards the optimal solution, we perform additional experiments with uniform and normal initial distributions. Fig. 7 shows the subsequent mean-field distributions in one episode after the algorithm reached a stable policy. We observe that M$^3$–UCRL is robust to the initial distribution and swiftly drives the mean-field distribution towards a steady state that maximizes the rewards over time.

