# OpenReview forum: "Efficient Model-Based Multi-Agent Mean-Field Reinforcement Learning"
_TMLR — Accepted by TMLR_

### Review · Reviewer_yvms · 2023-02-06

**Summary Of Contributions:**

This paper addresses the problem of learning an optimal control policy in cooperative mean-field games with unknown transition dynamics. It proposes a model-based reinforcement learning approach that is an adaptation of the UCRL algorithm to this specific setting. Especially, it provides an analysis of the proposed algorithm, which results in a $O( \sqrt{H^3 T})$ regret upper bound under smoothness and calibration assumptions over the transition dynamics. The paper further explains how the proposed algorithm can be implemented with GPs or neural networks retaining some of the theoretical guarantees. Finally, the algorithm is numerically validated in a continuous gridworld in which the rewards are the entropy of the mean-field distribution.

**Audience:**

Yes

**Broader Impact Concerns:**

This paper provides an essentially theoretical contribution. While the presented approaches might be employed in sensible applications in the future, I do not think that an impact statement is needed at this stage of development.

**Claims And Evidence:**

Yes

**Requested Changes:**

I believe that the current manuscript would need the following changes in order to reach the bar for acceptance:
1) An interpretation of the regret results in comparison to other (possibly model-free) approaches for mean-field games or against a lower bound;
2) A thorough discussion of the model assumptions, how much is the resulting setting expressive and what kind of applications can be modeled under these assumptions.

Other changes that can add value to this work are:
- A more substantial experimental analysis of the proposed algorithm in challenging domains and against some relevant baseline;
- A revised notation that draws a better link between the considered setting and the usual MDP setting.

**Strengths And Weaknesses:**

Strengths

- (Analysis) The paper provides a regret analysis for a typical optimistic RL procedure adapted to cooperative mean-field control, which is claimed to be novel in the paper (I am not familiar with the mean-field games literature, and unaware of previous comparable results);
- (Minor. Suggested application) The paper numerically validate the proposed algorithm in a maximum state entropy exploration setting, which suggests an interesting application domain for mean-field control, although this was considered in the literature (e.g., Geist et al., 2021);
- (Minor. Clarity) The paper is well-written and easy to follow.

Weaknesses

- (Interpretation of the regret) The paper does not compare the regret bound of the proposed algorithm neither to previous MFGs results nor to statistical limits of the settings, which makes hard to fully understand the value of the analysis;
- (Discussion of the model assumptions) The regret analysis is provided under two model assumptions (smoothness of the dynamics and calibration) that looks relatively strong. Especially, it is not clear from the paper how much expressive the setting is under those assumptions;
- (Minor. Numerical validation in toy domain) The paper stresses the practical upside of the procedure, but only provides a numerical validation in a simple domain, without comparing against any alternative approach.

This paper tackles an interesting multi-agent learning setting addressing sample efficiency and scalability through a mean-field game formulation. I have limited knowledge on the related literature, but I was not aware of a previous analysis of UCRL-based approaches in this setting. However, I think the paper is somewhat weak in terms of interpretation of the value of the provided analysis and discussion of the modeling assumptions (see comments below). If the paper could improve in those aspects (see requested changes), I believe it might be a valuable contribution within the TMLR journal.

Detailed Comments

(C1. Notation) The notation looks to be closer to previous works in mean-field games rather than RL, e.g., denoting the transition dynamics through a McKean-Vlasov stochastic process instead of MDP-like notation. This makes the paper slightly harder to process for an RL audience, and I would consider making a clearer connection between the proposed notation and usual MDP components.

(C2. Strength of the model assumptions) In the proposed setting, the transition dynamics is modeled through a Lipschitz function with additive noise. How expressive is the setting under this assumption? What kind of applications can actually model? Especially, it does not seem to include tabular domains in general, which might limit its applicability.

(C3. Model calibration) The calibration of the statistical model requires the existence of element-wise confidence bounds over then transition function. This seems to be relatively strong, especially if the environment admits states-distribution pairs that cannot be frequently reached with any policy.

(C4. Regret bound interpretation) The regret upper bound provided in the paper does not include any explicit dependence with the size of the problem (from the appendix, it appears to be linear in the state dimensions p, but the action dimensions q does not appear). Moreover, the paper does not compare the obtained result with previous works. Even if this might be the first paper tackling this specific setting, can the authors at least roughly compare their results with previous analyses in finite domains?

(C5. Regret lower bound) Without a lower bound setting a statistical barrier for this specific setting, it is hard to understand how far is the proposed algorithm from minimax optimality. Even without a lower bound, can the authors discuss the reported dependencies and whether they think the bound can be improved?

(C6. Experiments) The paper stresses the fact that the proposed approach has practical upside, especially in combination with GPs or neural networks. However, it only reports a numerical validation in a toy domain, and does not compare against alternative approaches for mean-field games or state entropy maximization (e.g., Hazan et al., Provably efficient maximum entropy exploration, 2019).

---

> ### Author Response · Authors · 2023-03-05
> **Response to the reviewer's comments**
>
> We would like to thank the reviewer for their effort to provide a thorough and constructive feedback on our paper. In the following, we would like to address their comments. We are happy to include further details on the discussed points in the final version of the paper.
>
> - **(C1. Notation)** We aim to use a notation that facilitates readability and the ease of understanding while following related works. However, if the reviewer has specific suggestions on how to improve the notation used in our paper, we are happy to consider them.
>
> - **(C2. Model calibration)** Assumption 3 (Calibrated Model) is satisfied for various model estimation techniques. In particular, Gaussian Process models satisfy this assumption when the unknown function $f$ has a bounded norm in the Reproducing Kernel Hilbert Space associated with the used kernel function and $\omega_{t,h}$ is $\sigma$-sub-Gaussian (See Assumption 5 and Lemma 7 in Appendix D). Additionally, Neural Network models such as Probabilistic and Deterministic Ensembles can be recalibrated to satisfy this assumption. We discuss this under the Practical Implementation section on the top of page 6.
> We note that Assumption 3 is not restrictive when certain inputs are not observed frequently. In this case, the epistemic uncertainty, $\sigma_t$, of the statistical model remains high. On the other hand, unreachable areas do not limit $M^3-UCRL$’s performance either since its goal is to maximize the social reward (Eq. (4)) and not to explore the whole state space. If a particular area is unreachable by any policy, it is also unreachable by the socially optimal policy, $\pi^*$, therefore, $M^3-UCRL$ does not have to explore it in order to achieve the optimal reward of the problem.
>
> - **(C4. Regret bound interpretation)** The complexity of the problem, including state and action space sizes, is captured by the $I_T$ (defined in Eq. (6)) term in the regret upper bound. For more complicated problems, this complexity measure will be higher and change the regret upper bound. In the specific case of Gaussian Process models, we show in Appendix D that the regret bound depends on the choice of the kernel and the related Maximum Information Gain. For the most commonly used kernels, the Maximum Information Gain is a function of the number of samples ($pHT$) and the dimensions of the input space $S \times A \times P(S)$. \
> In comparison to other model-based algorithms, M3-UCRL achieves $O(\sqrt{H^3 T I_T})$ which is equivalent to that of H-UCRL (Curi et al. (2020)) while H-MARL (Sessa et al. (2022)) archives $O(N^{H/2}\sqrt{H^3 T I_T})$. The additional factor of $O(N^{H/2})$ comes from the fact that H-MARL considers separate agents with individual action spaces while $M^3-UCRL$ uses a representative agent instead. Additionally, in Appendix E.4, we compare $M^3-UCRL$ to the model-free U2-MF-QL-FH algorithm which is the only comparable in the literature to the best of our knowledge. Other algorithms are considering infinite horizon problems and finding stationary solutions which is a significantly different concept, hence, comparison would be misleading. Our comparison shows that the other model-free approach fails to converge in 10 million episodes which is six orders of magnitude larger than the number of episodes $M^3-UCRL$ requires.
>
> - **(C5. Regret lower bound)** Our theoretical results show the general regret bound of $O(\sqrt{H^3 T I_T})$. The dependence on $T$ and $I_T$ are unavoidable in our opinion. In the simpler kernelized bandit setting using Gaussian Process models and bounded RKHS norm, $\sqrt{\gamma_T * T}$ has been shown to be near optimal (Lower Bounds on Regret for Noisy Gaussian Process Bandit Optimization, Scarlett et al. (2017)) for the Squared Exponential and Matern kernels. This closely relates to the single-agent problem with H=1, hence we expect that the term $\sqrt{T I_T}$ is necessary in our bound. The dependence on the time horizon $H$, however, could potentially be improved under more restrictive assumptions.
>
> - **(C6. Experiments)** Given the theoretical nature of the contribution of our paper, our experiment was meant to showcase the convergence of $M^3-UCRL$ and not as an exhaustive test of its application. Since we focus on the problem of Mean-Field Control, we provide a comparison to a model-free variant in Appendix E.4 but did not consider other state entropy maximization techniques as they seem out of scope given the focus of the paper. In particular, we chose this experiment to align with the recently published review of the field (Lauriere et al. (2022)) and encourage further adaptation of the problem for comparison purposes. However, we agree that other benchmark problems and challenges that could help the comparison of algorithms would be beneficial to the progress of Mean-field Games and Controls.

---

> > ### Comment · Reviewer_yvms · 2023-03-14
> > **Follow up**
> >
> > I want to thank the authors for their clarifications and report some follow-up comments.
> >
> > Regarding my first comment **(C1. Notation)**, I was not implying that the notation is not clear per se, but it is different from the common notation of theoretical RL papers. To help the latter audience access the paper, I would look for representative works in this field (e.g., Auer et al., Near-optimal regret bounds for reinforcement learning, 2010; Azar et al., Minimax regret bounds for reinforcement learning, 2017; Jin et al., Provably efficient reinforcement learning with linear function approximation, 2020) and try to connect the notation to their problem formulation and assumptions.
> >
> > **(C4. Regret bound interpretation)** I see that the factor $I_T$ depends on the complexity of the problem, but this does not seem to be easy to interpret, and I would rather see explicit dependencies on the size of the MDP.
> >
> > Can the authors also address the comment **(C2. Strength of the model assumption)**? Especially, how this relate to typical model assumptions in theoretical RL, such as tabular MDPs,  linear MDPs (Jin et al., 2020), linear-mixture MDPs (Ayoub et al., Model-based reinforcement learning with value-targeted regression, 2020), and others?

---

> > > ### Author Response · Authors · 2023-03-22
> > > **Follow up response**
> > >
> > > We kindly thank the reviewer the follow up comments and thorough evaluation of our work.
> > >
> > > Regarding **(C1. Notation)**, we appreciate the suggestions and will evaluate the notation changes suggested.
> > >
> > > **(C4. Regret bound interpretation)**: The dependency on the size of the MDP depends on the chosen calibrated model and its efficiency to estimate the underlying unknown function, therefore, it is hard to provide an explicit relationship between the two. In the case of Gaussian Processes, we show in Appendix D that $I_T$ relates to maximum information gain of the GP model which is specific to the chosen kernel. Afterall, the regret bound’s dependency on the size of the MDP is determined by the kernel with which the GP model satisfies Assumption 3 (Calibrated Model).
> > >
> > > **(C2. Strength of the model assumption)**: Accidentally, our initial response was missing the answer to this comment due to an editing mistake. Please find it below.
> > >
> > > Similar assumptions have been considered in the related model-based reinforcement learning literature, showcasing that it does not significantly limit the applicability of model-based approaches. For example, the problem of Linear-Quadratic Regulator (Abbasi-Yadkori & Szepesvari (2011)) fits this assumption if the transition matrix has a bounded norm, while Curi et al. (2020) and Sessa et al. (2022) assume similarly Lipschitz dynamics with additive noise.

---

### Review · Reviewer_6s4o · 2023-02-17

**Summary Of Contributions:**

This work presents a novel model-based reinforcement learning (MORL) approach for mean-field control (MFC) entitled Model-based Multi-agent Mean-field Upper Confidence RL (M$^3$– UCRL). It provides the first general regret bounds for model-based reinforcement learning for MFC, along with a practical parametrisation  that allows the use of gradient-optimisation based techniques to solve the problem.

**Audience:**

Yes

**Broader Impact Concerns:**

No concerns at this moment.

**Claims And Evidence:**

No

**Requested Changes:**

As far as I can tell, currently, the work follows the structure of the H-UCRL contribution, but I think it could benefit for a stronger positioning in the MFC literature. Following for example the survey of Laurière et al. (2022), you can describe/highlight in your setting details such as evolutive setting, cooperative games, social welfare optimisation.

To improve the clarity of the work, there are a couple of statements that should be strengthen throughout the work:
- "a novel mean-field type analysis" - concretely specify what is the the novelty of the analysis
- the sample-efficiency of M^3– UCRL. This property is also present in the title, however I do not find it is concretely and sufficiently supported in the work currently
- related to the previous point, statements such as "Our results show that M3–UCRL is capable of finding close-to-optimal policies within a few episodes, in contrast to model-free algorithms that require at least six orders of magnitude more samples." should also be supported by empirical evidence present in the main body of the paper. To this end, I suggest to include results from Appendix E4 in the main paper.

The theoretical analysis also includes bounds for GP models, however, as far as I can tell, there is no empirical evaluation that employs GPs. Would it be possible to extend the results with experiments that include the use of a GP model, perhaps on a different environment? (e.g., a congestion game).

Additionally, for the deep ensemble model, can you present more implementation details on the model calibration (or how Assumptions 3 and 4 are met), and how the posterior distribution over dynamical models is derived.

Finally, in my opinion, appendix A should also be part of the main paper.


**Strengths And Weaknesses:**

**Strengths:**

The work proposes a novel MORL approach for MFC, based on Hallucinated-Upper-Confidence Reinforcement Learning (H-UCRL), together with a general theoretical analysis on the regret bound. A practical implementation of M^3– UCRL is provided and evaluated in the exploration via entropy maximization problem.

**Weaknesses:**

The main body of the paper does not currently encompass all the necessary details for a clear and reproducible work, thus some restructuring is in order.
There are a few aspects and statements that require clarification.
The theoretical and empirical evaluation are not in full alignment.

(more details below).

---

> ### Author Response · Authors · 2023-03-05
> **Response to the reviewer's comments (1/2)**
>
> We would like to thank the reviewer for their effort to provide a thorough and constructive feedback on our paper. In the following, we would like to address their comments. We are happy to include further details on the discussed points in the final version of the paper.
>
> > *As far as I can tell, currently, the work follows the structure of the H-UCRL contribution, but I think it could benefit for a stronger positioning in the MFC literature. Following for example the survey of Laurière et al. (2022), you can describe/highlight in your setting details such as evolutive setting, cooperative games, social welfare optimisation.*
>
> We thank the reviewer for their remarks on the positioning of the paper in the MFC literature. We agree that aligning with the terminology in the recent survey (Lauriere et al. (2022)), e.g., evolutive setting, is beneficial and glad to provide further remarks on these points in the final version of the paper.
>
> > *To improve the clarity of the work, there are a couple of statements that should be strengthen throughout the work:*
>
> - > *"a novel mean-field type analysis" - concretely specify what is the the novelty of the analysis*
>
> As we highlight on the top of page 7 in the scratch of Theorem 1’s proof, the main contribution and novelty of our theoretical analysis is provided in Lemma 5 that bounds the Wasserstein-1 distance of the mean-field distributions under the true dynamics, $\mu_{t,h}$, and the approximated dynamics $\tilde{\mu}_{t,h}$. This is a non-trivial bound that is unique to the Mean-Field Control setting and the $M^3-UCRL$ algorithm.
>
> - > *the sample-efficiency of $M^3– UCRL$. This property is also present in the title, however I do not find it is concretely and sufficiently supported in the work currently*
>
> Our theoretical work shows that $M^3-UCRL$ obtains sublinear regret and achieves a similar bound than Curi et al. (2020) for the single-agent setting. (For further explanation on the bound, see our answer to reviewer yvms’ comment C5). We also demonstrate efficiency empirically by showing that a comparable model-free algorithm fails to converge in several millions of episodes while our approach requires only around 30. We will make sure to further clarify the interpretation of our regret bound and provide more details of the comparison in the main text of the final version.
>
>  - > related to the previous point, statements such as "Our results show that $M^3–UCRL$ is capable of finding close-to-optimal policies within a few episodes, in contrast to model-free algorithms that require at least six orders of magnitude more samples." should also be supported by empirical evidence present in the main body of the paper. To this end, I suggest to include results from Appendix E4 in the main paper.
>
> We agree that the comparison in Appendix E4 is an important result of our paper, however, we decided to defer the details to the appendix for brevity and ease of reading. Our aim is to present the $M^3-UCRL$ algorithm concisely in the main text and refer an interested reader to the appendix for additional details. The sentence highlighted in the review already summarizes the result of the comparison and moving additional results such as Figure 5 would require further paragraphs of explanation of the comparison. However, we are happy to provide further remarks in Section 5 on why the model-free algorithm, $U2-MF-QL-FH$, fails to converge such as large state and action spaces, deterministic policy, and challenges in exploration. We hope this would be sufficient evidence to support our claim.

---

> ### Author Response · Authors · 2023-03-05
> **Response to the reviewer's comments (2/2)**
>
> > *The theoretical analysis also includes bounds for GP models, however, as far as I can tell, there is no empirical evaluation that employs GPs. Would it be possible to extend the results with experiments that include the use of a GP model, perhaps on a different environment? (e.g., a congestion game).*
>
> During the research, we have conducted experiments in the Swarm motion environment used by Carmona et al. (2019b) and Elie et al. (2020) with Gaussian Process models. To provide further evidence and examples of how to use GPs, we could include those results in the Appendix of the paper’s final version.
>
> > *Additionally, for the deep ensemble model, can you present more implementation details on the model calibration (or how Assumptions 3 and 4 are met), and how the posterior distribution over dynamical models is derived.*
>
> We can provide further implementation details in the Appendix. Our implementation follows Lakshminarayanan et al. (2017) using 10 feed-forward neural networks each with one hidden layer of size 32 and leaky ReLu activation. As described in the paper, we fit each Neural Network by minimizing the negative log-likelihood under the assumption of heteroscedastic Gaussian distribution, therefore, the output of each network is both the estimated mean and variance for each sample. Additionally, we also implemented the adversarial training procedure from Lakshminarayanan et al. (2017) to smooth the networks’ output and increase their robustness. The posterior mean and epistemic variance are calculated as the empirical statistics of the  Neural Networks’ estimated mean values. We found that this statistical model already performs sufficiently well, therefore, we do not implement additional recalibration procedures such as Kuleshov et al (2018).
>
> > *Finally, in my opinion, appendix A should also be part of the main paper.*
>
> If the action editor agrees that this would enhance the quality of the paper, we are happy to move Appendix A with the pseudo-code of $M^3-UCRL$ to Section 3.

---

> > ### Comment · Reviewer_6s4o · 2023-04-18
> > **Thank you for the clarifications**
> >
> > I thank the authors for all the clarifications and I think the revised version addressed the raised concerns and improves the clarity of the work. I maintain a positive evaluation.

---

### Review · Reviewer_SDgn · 2023-02-20

**Summary Of Contributions:**

 The authors study multi-agent reinforcement learning, wherein they tackle the mean-field control problem. This formulation assumes an asymptotically infinite population of agents interacting with a common environment and aiming to collaboratively maximize a collective reward. A model-based algorithm is proposed, for which regret bounds are provided. The algorithm is tested in numerical simulations.

**Audience:**

Yes

**Claims And Evidence:**

Yes

**Requested Changes:**

A better experimental section would improve the manuscript. Nonetheless, I found this work well-written and deserving of publication.


**Strengths And Weaknesses:**

The problem studied in this work is of interest to the community. The manuscript is well written and the ideas introduced, while simple, appear to be novel and work sufficiently well. The supporting theory introduced is treated with rigor and its analysis appears to be correct.

Overall, the biggest weakness of the manuscript is in its Experiments section. The problem considered is quite simple, even though it has been used previously as a comparison benchmark in Laurière et al., 2022. However, the case considered previously is for discrete state-action spaces, while the authors consider the continuous case, making performance comparisons difficult. The manuscript would benefit from either more complex simulations and/or the application of the authors method to the discrete case to better compare it with the previous state of the art.

---

> ### Author Response · Authors · 2023-03-05
> **Response to the reviewer's comments**
>
> We thank the reviewer for the supportive review. While we tried to compare our results to the survey and a model-free algorithm, as pointed out, the comparison is difficult due to the difference in the state-action spaces. We hope that by formulating the problem in continuous spaces and conducting experiments, we encourage further its adoption as a benchmark. We believe that standard experiments and further challenges would be beneficial to the progress of the field.

---

### Author Response · Authors · 2023-04-17
**Revision update**

Dear Reviewers,

We uploaded a revised version of the paper following our constructive discussions.
The following changes have been made:
* Made a remark at the end of Section 2 (Problem Statement) that compares our problem setting to Lauriere et al. (2022)
* Moved the pseudoalgorithm to Section 3 (The $M^3-UCRL$ Algorithm) from the Appendix
* Elaborated on the regret bound's interpretation after Theorem 1
* Added the swarm motion experiments from previous submissions to the Appendix and refer to it in Section 5 (Experiments)
* Added further comments on the model-free comparison to the end of Section 5 (Experiments)
A difference file (revision_difference_highlights.pdf) highlighting the changes has been added to the supplementary material. Text highlighted in blue shows new text in the paper and text highlighted in red shows removed text.

Kind regards, Authors

---

> ### Comment · Reviewer_yvms · 2023-04-18
> **Thank you for the revision**
>
> I want to thank the authors for updating the paper to accomodate reviewers' suggestions. I think the changes are improving the clarity of the paper overall.
>
> Best,
> Reviewer yvms

---

### Decision · Action_Editors · 2023-03-27

**Recommendation:** Accept as is

**Comment:**

Reviewers find the work novel and interesting, while offering a number of detailed suggestions to improve the paper. Most of them are about improving clarity of the positioning, limitations of assumptions, technical novelty, among others. The authors provided a revised version that addresses these questions. We are happy to recommend acceptance.

**Audience:**

This work is relevant to the communities of multi-agent RL, mean-field control, and online learning (exploration vs exploitation).

**Claims And Evidence:**

This work presents a model-based RL algorithm (M3-UCRL) for mean-field control (MFC), provides regret bounds, demonstrates how to implement the algorithm with function approximation like GPs and NNs, and includes proof-of-concept numerical experiments. These claims are supported by analysis and simulations. Reviewers find the evidence to be solid, with suggestions on how to strengthen the constributions.

---

> ### Author Response · Authors · 2023-05-01
> **Camera-ready version**
>
> Dear Editor,
>
> Thank you for your time and effort managing the review of this work and we are pleased to receive a positive decision.
> The camera-ready version has been uploaded.
>
> Kind regards,
> Authors